# Systematic decomposition of sequence determinants governing CRISPR/Cas9 specificity

Rongjie Fu[1,8], Wei He[1,8], Jinzhuang Dou[1], Oscar D. Villarreal[1], Ella Bedford[1], Helen Wang[1], Connie Hou[1], Liang Zhang[1], Yalong Wang[1], Dacheng Ma[2], Yiwen Chen [3], Xue Gao [2,4,5], Martin Depken [6] & Han Xu [1,3,7✉]

The specificity of CRISPR/Cas9 genome editing is largely determined by the sequences of guide RNA (gRNA) and the targeted DNA, yet the sequence-dependent rules underlying off-target effects are not fully understood. To systematically explore the sequence determinants governing CRISPR/Cas9 specificity, here we describe a dual-target system to measure the relative cleavage rate between off- and on-target sequences (off-on ratios) of 1902 gRNAs on 13,314 synthetic target sequences, and reveal a set of sequence rules involving 2 factors in off-targeting: 1) a guide-intrinsic mismatch tolerance (GMT) independent of the mismatch context; 2) an "epistasis-like" combinatorial effect of multiple mismatches, which are associated with the free-energy landscape in R-loop formation and are explainable by a multi-state kinetic model. These sequence rules lead to the development of MOFF, a model-based predictor of Cas9-mediated off-target effects. Moreover, the "epistasis-like" combinatorial effect suggests a strategy of allele-specific genome editing using mismatched guides. With the aid of MOFF prediction, this strategy significantly improves the selectivity and expands the application domain of Cas9-based allele-specific editing, as tested in a high-throughput allele-editing screen on 18 cancer hotspot mutations.

[1] Department of Epigenetics and Molecular Carcinogenesis, The University of Texas MD Anderson Cancer Center, Smithville, TX 78957, USA. [2] Department of Chemical and Biomolecular Engineering, Rice University, Houston, TX 77005, USA. [3] Department of Bioinformatics and Computational Biology, The University of Texas MD Anderson Cancer Center, Houston, TX 77030, USA. [4] Department of Chemistry, Rice University, Houston, TX 77005, USA. [5] Department of Bioengineering, Rice University, Houston, TX 77005, USA. [6] Kavli Institute of NanoScience and Department of BionanoScience, Delft University of Technology, Delft 2629HZ, the Netherlands. [7] The Center for Cancer Epigenetics, The University of Texas MD Anderson Cancer Center, Houston, TX 77030, USA. [8] These authors contributed equally: Rongjie Fu, Wei He. ✉email: hxu4@mdanderson.org

CRISPR/Cas9 technology has been widely used for genome editing and is currently being tested as a therapeutic in clinical trials[1–3]. However, the risk of Cas9 cleaving sequences with high similarity to the targeted DNA has raised critical concerns in its scientific and clinical applications[4–8]. Thus, it is crucial to diminish off-target effects in CRISPR/Cas9 genome editing.

A number of experimental techniques, including high-throughput screens[9–11] and various genome-wide detection approaches[12–19], have been developed to quantitatively assess the off-target effects. These approaches have facilitated the exploration of rules governing the specificity of CRISPR/Cas9, and it is widely accepted that the off-target cleavage by Cas9 is dependent on the positions and nucleotide contexts of mismatches between crRNA and targeted protospacer[9,11,20]. Nevertheless, these mismatch-dependent rules only partially account for the observed off-target effects. For example, recent studies revealed a guide-intrinsic mismatch tolerance (GMT) in addition to the impact of specific mismatches[21,22]. While the observations of GMT suggest the possibility to identify the highly specific gRNA regardless of genetic context, the sequence determinants underlying GMT is unclear. Over 95% of the detected genomic off-target sites harbor two or more mismatches with respect to the crRNA sequences[23], but the combinatorial effect of multiple mismatches is not well understood due to the lack of adequate experimental data that allow for quantitative separation of the combinatorial effect from that of individual mismatches.

Although experimental techniques are capable of quantifying Cas9 off-target activities, in silico prediction of the off-target effects remains the most efficient and cost-effective method for designing and optimizing CRISPR-based applications. The advancement of machine learning approaches has fueled the progressive improvement of off-target prediction over the past several years[24–29]. Moreover, biophysical modeling has provided new insights into the prediction of off-targeting from a bottom-up perspective[30–32]. Notwithstanding this progress, the power for computational prediction of CRISPR/Cas9 off-target effects remains limited as large-scale training datasets covering more gRNAs and an enhanced understanding of sequence-dependent rules are needed.

In the present study, we devise a high-throughput synthetic system with a dual-target design to measure the relative cleavage rate between off- and on-target sequences (off-on ratios) of ~2,000 gRNAs. This system shows lower variance among experimental replicates and better assessment of the genomic off-targets compared to the previous single-target system. The large number of gRNAs and customized library design facilitate the mining of a more comprehensive set of rules underlying CRISPR/Cas9 specificity, which lead to the development of an improved off-target prediction tool and an optimized strategy for allele-specific genome editing.

## Results

**High-throughput assessment of off-on ratios using a synthetic dual-target system.** Recently, paired gRNA-target systems have been developed for the high-throughput assessment of CRISPR/Cas9 genome editing outcomes[33–35]. These systems include a gRNA expression cassette together with a target sequence flanked by barcodes that are uniquely mapped to the gRNA sequence (Fig. 1a). Combined with high-throughput oligonucleotide synthesis and lentiviral delivery, the variable context of target sequences allows flexible design of the experiments to detect Cas9-mediated editing events associated with a large number of gRNAs. To facilitate a direct comparison of the off- and on-target effects, we modified the paired gRNA-target system by introducing a dual-target sequence that contains two 23-bp PAM-endowed target sequences arranged

in tandem, corresponding to an off-target (left) and an on-target (right). These two targets are separated by an optimized 15-nt linker sequence and are surrounded by a 10-nt barcode 1 at the 5′ end and a 15-nt barcode 2 at the 3′ end (Fig. 1b). Since the off- and on-targets are integrated to the same genomic locus and are PCR-amplified together, the on-target cleavage rate acts as an internal control for the normalization against confounding factors in the experiment. Compared to the single-target design without the use of internal normalizations, the dual-target design is expected to reduce the experimental variations and biases for accurate measurement of off-on ratios.

To explore the editing outcomes mediated by different cleavage events at almost identical tandem targets, we designed control dual-target sequences to represent four combinations of cleavage events (no cleavage, left, right, and both), where the cleavage can be turned off at a specific target by the replacement of the "NGG" PAM sequences with "NTT" (Fig. 1c). We first tested two gRNAs associated with distinct repair mechanisms upon double-strand breaks (Fig. 1d and Supplementary Fig. 1). In addition to anticipated small indels, large deletions (>30-nt) are enriched when cleavage occurs at both targets (NGG + NGG) or the left target alone (NGG + NTT). The latter is likely due to the similarity of the two target sequences that induces long-range resection via microhomology-mediated end joining (MMEJ). These observations are consistent between the two gRNAs, suggesting a general cleavage-editing model as demonstrated in Fig. 1e. We further extended the analysis on 276 control gRNA-target pairs using a high-throughput pooled screen. Computational analysis of the Cas9-mediated mutational profiles at the control sequences rendered a matrix that represents the probability of editing outcomes conditional on cleavage types, as well as the background noise rates measured from the "no cleavage" sequences (Fig. 1f). This matrix enabled subsequent inference of the cleavage frequency and off-on ratio from the observation of editing outcomes (see Methods section for details).

To evaluate the dual-target approach, we designed a library that includes 35 benchmark gRNAs collected from previous in vitro or in vivo studies, corresponding to 296 reported off-target sites (positive controls) and 295 similar genomic sequences that are never detected to be off-targets (negative controls)[1,12,17,36]. Additionally, the library includes 328 random gRNAs corresponding to 2261 off-targets harboring 1–3 mismatches (Supplementary Data 1). For a systematic comparison, we also generated a single-target library (Fig. 1a and Supplementary Data 2) that includes the same sets of off- and on-targets. Both lentiviral libraries were transduced to HEK293T cells expressing doxycycline-inducible Cas9 (HEK293T-iCas9) for the assessment of off-on ratios, following almost identical experimental configurations (Supplementary Fig. 2). As expected, the dual-target system significantly improved the correlations of the measured off-on ratios between biological replicates or different barcode sets (Fig. 1g and Supplementary Fig. 3). Comparison of the positive and negative control targets suggests a dynamic range of ~0.02–1.0 in the measurement of off-on ratios for the dual-target system (Fig. 1h). On the positive controls, the off-on ratios detected by the dual-target system showed higher consistency with the measures from in vitro GUIDE-seq[12] (Fig. 1i and Supplementary Fig. 4a) and an in vivo study on mouse embryos using whole-genome sequencing (WGS)[36] (Fig. 1j and Supplementary Fig. 4b) as compared to the single-target system. Overall, these head-to-head comparisons confirmed the rationale and advantage of the dual-target system that improves the accuracy of off-on ratio measurements via internal normalization using on-target controls.

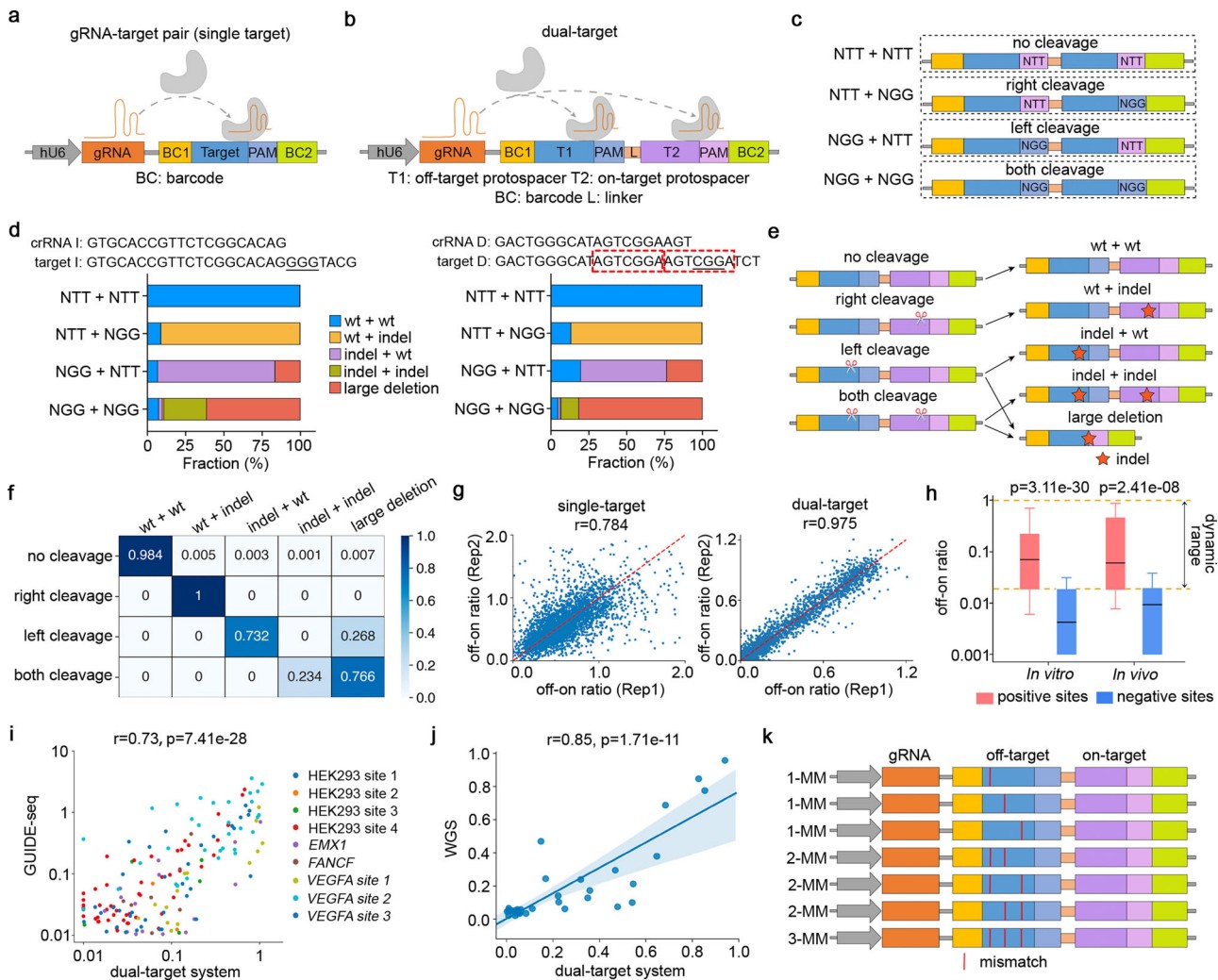

**Fig. 1 Assessment of off-on ratios using a synthetic dual-target system. a, b** A schematic representation of paired gRNA-target design using **a** single-target or **b** dual-target sequence. **c** The design of control dual-target sequences corresponding to 4 cleavage types, where the left and right protospacers are identical and the cleavage events are turned on or off by "NGG" or "NTT" PAM sequences, respectively. **d** Bar charts showing the fractions of five types of editing outcomes in relation to the 4 cleavage types. Left: a gRNA associated with dominating non-homologous end joining (NHEJ); right: a gRNA associated with dominating microhomology-mediated end joining (MMEJ), where the microhomology sequences are highlighted in red rectangles. **e** A demonstration of the cleavage-editing model of the dual-target system. **f** A heatmap showing the cleavage-editing transition matrix computed from mutational profiles of 276 control dual-target sequences. **g** Scatter plots showing the correlations of off-on ratios between biological replicates using single- or dual-target systems. **h** A box plot showing the distributions of the off-on ratios estimated by the dual-target system, corresponding to the 32 filtered benchmark gRNAs and 480 target sequences. Positive: reported off-target sequences; Negative: unreported genomic sequences with <5 mismatches relative to crRNA. The data represent $n = 204$ positive, 213 negative (in vitro), and $n = 66$ positive, 52 negative (in vivo) gRNA-target pairs. The box plot displays a median line, interquartile range boxes and min to max whiskers. The $p$-values were calculated using two-tailed Manny–Whitney $U$ test. **i, j** Scatter plots showing the correlations between the off-on ratios estimated from the dual-target system and **i** GUIDE-seq or **j** WGS, at the reported genomic off-target sequences. The $p$-values were calculated using Pearson correlation test. The shadow represents the 95% confidence interval. **k** A demonstration of the off-target sequence design in the high-throughput experiments using the dual-target synthetic system, where the mismatches in the 2-MM and 3-MM sequences are the combinations of the mismatches in the 1-MM sequences. Source data for Fig. 1d, g–j are provided in the Source Data file.

With the proven robustness of our system, we measured the off-on ratios of 1902 gRNAs on 13,314 synthetic target sequences (Supplementary Data 1). We designed 7 off-target sequences for each gRNA, including 3 targets with 1 mismatch (1-MM), 3 with 2 mismatches (2-MM), and 1 with 3 mismatches (3-MM). The mismatches in the 2-MM and 3-MM sequences are the combinations of the mismatches in 1-MM sequences, as exemplified in Fig. 1k. These settings allow systematic decomposition of the estimated off-on ratios into single-mismatch effect, combinatorial effect, and GMT.

**Sequence determinants of guide-intrinsic mismatch tolerance.** To confirm the existence of the GMT effect, we first estimated the indel rates of 6 gRNAs at synthetic targets harboring 0 to 6 mismatches (Supplementary Data 3). While the on-target indel rates are largely consistent for all the gRNAs, we found that 4 gRNAs were associated with high indel rates at the 1-MM sequences and the other 2 showed obvious mismatch intolerance (Fig. 2a and Supplementary Fig. 5). This guide-intrinsic effect was also observed at the 2-MM and 3-MM sequences regardless of various mismatch contexts (Fig. 2b). To quantitatively estimate

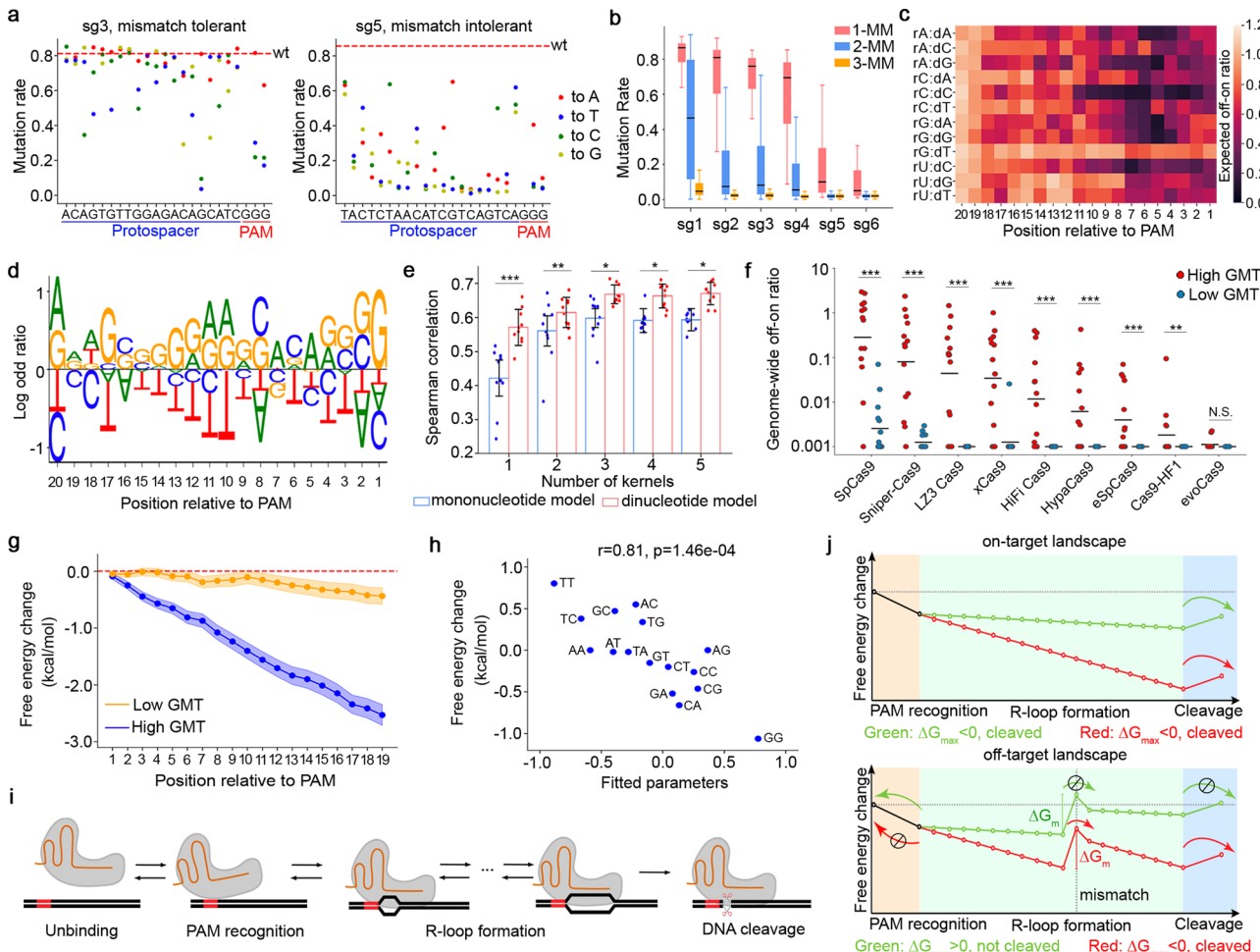

**Fig. 2 The guide-intrinsic mismatch tolerance (GMT) and its underlying biophysical mechanism. a** The mutation rates at 1-mismatch target sequences of two example gRNAs that are associated with high GMT (left) and low GMT (right). Each dot corresponds to a specific mismatched target. The red dashed line represents the mutation rate at perfectly matched target. **b** A box plot showing the distributions of off-on ratios of 6 gRNAs at synthetic single targets harboring 1-, 2-, or 3-mismatches. The data represent $n = 66$ 1-MM, 2,079 2-MM, and 100 3-MM gRNA-target pairs for each gRNA. The box plot displays a median line, interquartile range boxes and min to max whiskers. **c** A heatmap showing the mismatch-dependent effect conditioned on the position and nucleotide context of the mismatch. rX:dY represents a mismatch where a nucleotide X in crRNA is paired with a nucleotide Y in the complementary target DNA. **d** A sequence logo demonstrating the log-odds ratios of nucleotide frequency between gRNAs with high GMT (top 25%) and low GMT (bottom 25%). **e** Performance comparison of the mononucleotide and dinucleotide models in the prediction of GMT, with various numbers of convolutional kernels. Each dot represents an iteration of 10-fold cross-validation. *$p$<0.05, **$p$ < 0.01, ***$p$ < 0.001. The $p$-values were calculated using two-tailed $t$-test. Data ($n = 10$) are presented as mean values ±SD. The exact $p$-values from left to right are: 1.8e-04, 9.6e-03, 0.011, 0.017, 0.027. **f** A dot plot showing the specificity of the gRNAs predicted to be of high GMT (top 25%, $n = 15$) or low GMT (bottom 25%, $n = 15$), across a panel of Cas9 variants. The gRNA specificity is measured as the ratio of genome-wide off-target read counts to on-target read counts in TTISS experiments[18]. **$p$ < 0.01, ***$p$ < 0.001. The $p$-values were calculated using two-tailed Manny-Whitney $U$ test. The exact $p$-values from left to right are: 4.72e-05,1.68e-05, 3.32e-05, 1.22e-04, 9.75e-05, 6.93e-04, 6.93e-04, 0.019, 0.08. **g** The average free-energy landscapes of the gRNAs are associated with high GMT (top 25%) or low GMT (bottom 25%). The shadows represent standard error for 95% confidence interval. **h** A scatter plot showing the correlation between the fitted dinucleotide parameters in the prediction model and the differences of base-stacking energy between DNA-DNA and RNA-DNA hybridizations[31]. The dinucleotide parameters reflect the contributions of dinucleotides to GMT prediction. To avoid confounded interpretation, the parameters shown in the plot were fitted using a single kernel model. The $p$-value was calculated using Pearson correlation test. **i** A schematic plot illustrating the kinetic transition of the states during Cas9-mediated target editing. **j** An illustration of the free-energy landscapes of a high-GMT gRNA (red) and a low-GMT gRNA (green) at an on-target (top panel) or a 1-MM off-target (bottom panel) sites. The mismatch introduces an energy barrier ($\Delta G_m$) during R-loop formation. The probability of overcoming the energy barrier is determined by its size relative to the barrier to unbinding, $\Delta G_{max}$. The stop symbol represents the repressed direction of the reaction. The arrow represents the direction of the reaction. Source data for Fig. 2c–h are provided in the Source Data file.

the GMT effects of the gRNAs in the large-scale dataset, we used a gradient descent algorithm to decompose the off-on ratios at the 1-MM sequences into (i) mismatch-dependent effects determined by the position and nucleotide context of the mismatches, and (ii) the GMT effects independent of the mismatch contexts (see Methods section for details and Supplementary Data 4). The estimated mismatch-dependent effect, as demonstrated by a

matrix in Fig. 2c, is highly consistent with a previous report based on CRISPR/Cas9 functional screens[9] ($r = 0.86$, Supplementary Fig. 6), supporting the robustness of the decomposition.

Comparing the nucleotide frequency of the protospacer sequences associated with high or low GMT effect, we observed an enrichment of guanine and depletion of thymine in the high-GMT protospacers (Fig. 2d). Unlike the on-target activity that is

mainly dependent on the context of the nucleotides proximal to PAM[37], the GMT effect relies on the sequences in both the seed and non-seed regions. Further analysis showed only a moderate correlation between the on-target activity and the GMT effect ($r = 0.44$, Supplementary Fig. 7), suggesting that it is possible to select gRNAs that are associated with both high on-target activity and low GMT. To predict GMT from a sequence, we tested two convolutional neural network (CNN) regression models corresponding to mono- and di-nucleotide kernels (Supplementary Fig. 8). Cross-validation showed that the dinucleotide model achieved the optimal performance (Fig. 2e). Tested on independent TTISS[18] and CHANGE-seq[19] datasets, the predicted GMT score is significantly associated with the specificity of the gRNAs on the vast majority of Cas9 variants (Fig. 2f and Supplementary Fig. 9).

Several studies have confirmed that R-loop formation is required for CRISPR/Cas9-mediated DNA cleavage[38,39]. In principle, the progression of R-loop formation can be driven by free-energy transactions involved in substituting DNA-DNA duplex interactions for RNA-DNA hybrid interactions in the context of the Cas9 protein. The sequence dependence of this free energy should be largely set by the base-pairing energies, particularly by the base-stacking energy[30–32]. Comparing the GMT effect with the difference of cumulative base-stacking energy between DNA-DNA and RNA-DNA hybridization, we found that the high-GMT and low-GMT gRNAs are associated with distinct free-energy landscapes (Fig. 2g). Moreover, we observed a strong correlation between the dinucleotide-specific base-stacking energy and the weights of dinucleotides that contribute to GMT prediction (Fig. 2h), suggesting that the free-energy change accounts for a majority of sequence-dependent GMT effect. Next, we asked why those high-GMT gRNAs are mismatch tolerant. We considered a recent kinetic model in which the process of Cas9-mediated cleavage is a sequential transition among the states of unbinding, PAM recognition, R-loop formation, and cleavage[40] (Fig. 2i). A mismatch causes an energy barrier that could halt R-loop progression, and the probability of unbinding before cleavage is determined by the relative height ($\triangle G_{max}$) of the kinetic barriers. If $\triangle G_{max} < 0$, the free-energy barrier to cleavage is lower than that to unbinding, and cleavage is the most likely outcome; if $\triangle G_{max} > 0$, the free-energy barrier to cleavage is higher than that to unbinding, the R-loop formation is kinetically blocked, and unbinding is the most likely outcome. As illustrated in Fig. 2j, the difference of free-energy landscapes intrinsic to high-GMT or low-GMT gRNAs leads to different levels of $\triangle G_{max}$, thus resulting in a high or low probability of reaching the final cleavage state, respectively.

**Combinatorial effect of multiple mismatches.** In a simplified theoretical model, the combinatorial effect of two or more mismatches is assumed to be "marginally independent"[9]. That is, the tolerance to a combination of mismatches, in terms of off-on ratio, is taken to equal the multiplied tolerances for the individual mismatches. To test this model, we compared the observed off-on ratios of 2-MM targets to the expected off-on ratios computed from 1-MM targets. The marginally independent model provides an upper bound of the combinatorial effects, but many combinations are associated with much lower off-on ratios (Fig. 3a), suggesting an "epistasis-like" effect. Consistently, the CFD score[9], a benchmark method based on the marginal independence model, overestimates the off-target effects at genomic targets harboring multiple mismatches (Fig. 3b). The "epistasis-like" combinatorial effect is explainable by the above-mentioned kinetic model[40], where the combination of two tolerated

mismatches results in a higher energy barrier that blocks R-loop progression (Fig. 3c).

To quantitatively model the combinatorial effect, we introduced a parameter $\delta_{ij}$ to represent the ratio of the observed and expected effects when two mismatches occur at the $i$th and the $j$th nucleotide with respect to the PAM. A smaller value of $\delta_{ij}$ indicates a stronger "epistasis-like" effect. The maximum likelihood estimates of $\delta_{ij}$ are shown in a heatmap in Fig. 3d. Additionally, we computed a relative co-occurrence score (RCS) that represents the observed frequency of two mismatches relative to random expectation, based on 96,555 genomic off-target sites detected by CHANGE-seq[19] (Fig. 3e). Cross-referencing $\delta_{ij}$ and RCS, we derived three reproducible rules of the combinatorial effect. First, the combinatorial factor $\delta_{ij}$ is smaller when both mismatches occur in the seed region (<10-nt from PAM) and is greater when one of the mismatches occurs at the PAM-distal 19th or 20th nucleotide. Second, the "epistasis-like" effect is dependent on the distance between the two mismatches, where a minimal $\delta_{ij}$ is observed when the distance is between 1 and 6 (Fig. 3f). Third, a greater $\delta_{ij}$ is associated with adjacent mismatches (distance = 0) that correspond to a 2-nt bubble in RNA-DNA hybridization, as compared to two separated proximal bubbles of 1-nt in size. Of note, all three revealed rules can be recapitulated using the kinetic model, suggesting an underlying biophysical mechanism (Fig. 3g). In extension to the third rule, we further asked if the number of bubbles is associated with the combinatorial effect given a fixed number of mismatches. We examined the genomic sequences harboring 4–6 mismatches with respect to 109 gRNAs in the CHANGE-seq dataset[19] and found that the sequences with a smaller number of bubbles are more likely to be detected as off-target sites (Fig. 3h and Supplementary Fig. 10). Consistently, those sequences with fewer bubbles are associated with lower energy barriers computed from the cumulative dinucleotide base-stacking energy changes, providing a biophysical explanation of the "bubble rule" (Fig. 3i).

**Predicting off-target effect and guide specificity with MOFF.** With the observations of GMT and the combinatorial effect of multiple mismatches, we sought to combine these sequence-dependent rules for in silico CRISPR/Cas9 off-target evaluation. We developed MOFF, a model-based off-target predictor that includes three components corresponding to the multiplication of individual mismatch effect (IME), the combinatorial effect (CE), and the GMT effect (Fig. 4a). Given a gRNA and an off-target sequence, we defined a MOFF-target score to be the logarithm of the expected off-on ratio as follows:

$$S_{MOFF} = \sum_{i=1}^{k} \log(s_i) + \frac{2}{k} \sum_{i=1}^{k} \sum_{j=1}^{i} \log\left(\delta_{ij}\right) + k \log\left(s_{GMT}\right)$$

where $k$ is the number of mismatches, $s_i$ is the effect of the $i$th mismatch computed from the matrix in Fig. 2c, $\delta_{ij}$ is the pairwise combinatorial effect with respect to the positions of the $i$th and $j$th mismatches as demonstrated in Fig. 3d, and $s_{GMT}$ is the GMT effect estimated from the dinucleotide CNN regression model (see Methods section for a detailed explanation of MOFF-target score). To facilitate the genome-wide assessment of the specificity of a given gRNA, we also defined a MOFF-aggregate score which is the logarithm of the sum of predicted off-on ratios for all genomic sequences harboring up to six mismatches.

We evaluated the performance of MOFF-target and MOFF-aggregate using three independent datasets generated by the platforms of GUIDE-seq[12], TTISS[18], and CHANGE-seq[19]. We reasoned that the classification of off-target and untargeted sites is highly dependent on the sensitivity of each platform, where the

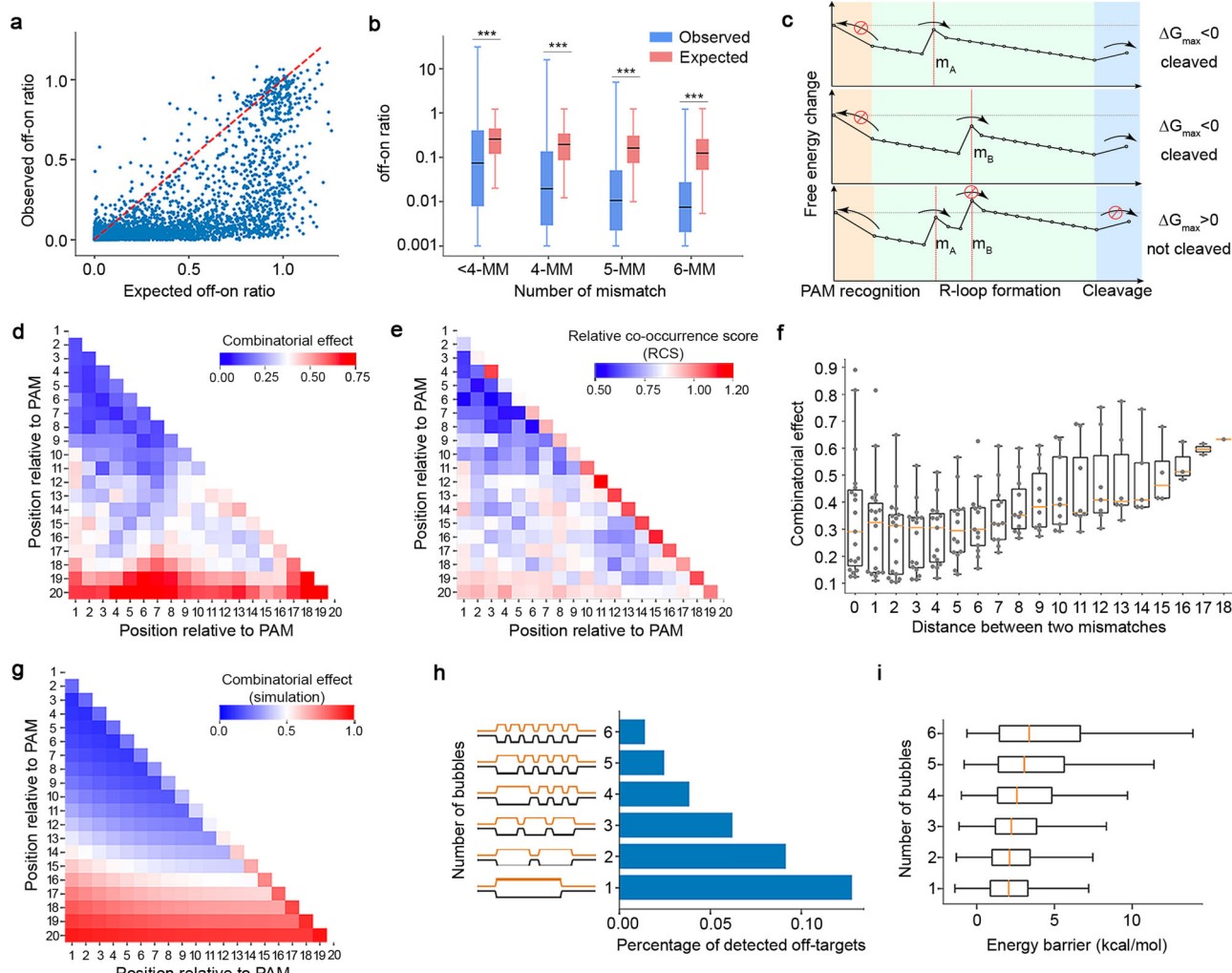

**Fig. 3 The combinatorial effect of multiple mismatches. a** A scatter plot showing the "epistasis-like" combinatorial effect of two mismatches. Each dot corresponds to a 2-MM target sequence. X-axis: the expected off-on ratio of the 2-MM target based on the marginal independence model. Y-axis: the observed off-on ratio at the 2-MM target. The red dashed line represents X = Y. **b** A box plot comparing the distributions of observed off-on ratios and predicted off-on ratios by CFD. Data were retrieved from published CHANGE-seq dataset on 109 gRNAs and 96,555 off-target sites[19]. ***$p$ < 0.001. The data represent $n$ = 1948 <4-MM, 11,893 4-MM, 34,709 5-MM, and 71,774 6-MM gRNA-target pairs in CHANGE-seq dataset. The $p$-values were calculated using two-tailed Manny–Whitney $U$ test. All the $p$-values are <1.0e-06. The box plot displays a median line, interquartile range boxes, and min to max whiskers. **c** An illustration of the free-energy landscapes at target sequences with tolerated single mismatches: $m_A$ and $m_B$ (upper and middle panels), as compared to the landscape at the sequence with the combination of these two mismatches (bottom panel). The red stop symbol represents the repressed direction of the reaction. The arrow represents the direction of the reaction. **d**, **e** Heatmaps showing **d** the position-dependent combinatorial effect ($\delta_{ij}$) of two mismatches, and **e** relative co-occurrence score (RCS) that represents the observed frequency of two mismatches relative to random expectation at off-target sites detected by CHANGE-seq. **f** A box plot showing the distributions of combinatorial effects conditioned on the distance between two mismatches. A distance of zero corresponds to adjacent mismatches that form a bubble of 2-nt in size. The data represent $n$ = 19, 18, 17, 16, 15, 14, 13, 12, 11, 10, 9, 8, 7, 6, 5, 4, 3, 2, and 1 situations where the distance between two mismatches is from 0 to 18. The box plot displays a median line, interquartile range boxes, and min to max whiskers. **g** A heatmap showing the expected combinatorial effect derived from the multi-state kinetic model[40]. For illustration we here show the case where the gain in energy due to PAM binding is taken to be 5 $k_BT$, the gain per correctly matched hybrid base pair is 0.25 $k_BT$, and the cost of a mismatch is 4 $k_BT$ if it is isolated, but only 3 $k_BT$ if it is added to an existing bubble. **h** A bar chart showing the percentage of off-target sites detected by CHANGE-seq. The off-target sequences are associated with six mismatches and are categorized based on the number of bubbles in DNA-RNA hybridization. **i** A box plot showing the distributions of cumulative energy barriers corresponding to 6-MM target sequences categorized based on the number of bubbles. The distributions were estimated from 1,000,000 random gRNA-target pairs, $n$ = 91, 4894, 50,342, 171,587, 208,612, and 76,779 gRNA-target pairs that harbor 1–6 bubbles. The energy barriers were computed based on dinucleotide stacking energy parameters[31]. The box plot displays a median line, interquartile range boxes and min to max whiskers. Source data for Fig. 3a, b, d–h are provided in the Source Data file.

off-target effects take continuous values that may differ by several orders of magnitude. Thus, we adopted the Spearman correlation in log-scale for quantitative evaluations. We compared the performance of MOFF-target and MOFF-aggregate to 5 off-target prediction methods, including the benchmark CFD score[9] and its improved version Elevation[24], two recent deep learning-based methods CNN_std[27] and CRISPR-Net[29], and an energy-based model CRISPRoff[31] (Fig. 4b-c). Among them, three machine learning-based methods (Elevation, CNN_std, and CRISPR-Net) achieved good predictive power on their training dataset generated by GUIDE-seq, but the performances degraded when tested on datasets from the other two platforms, suggesting

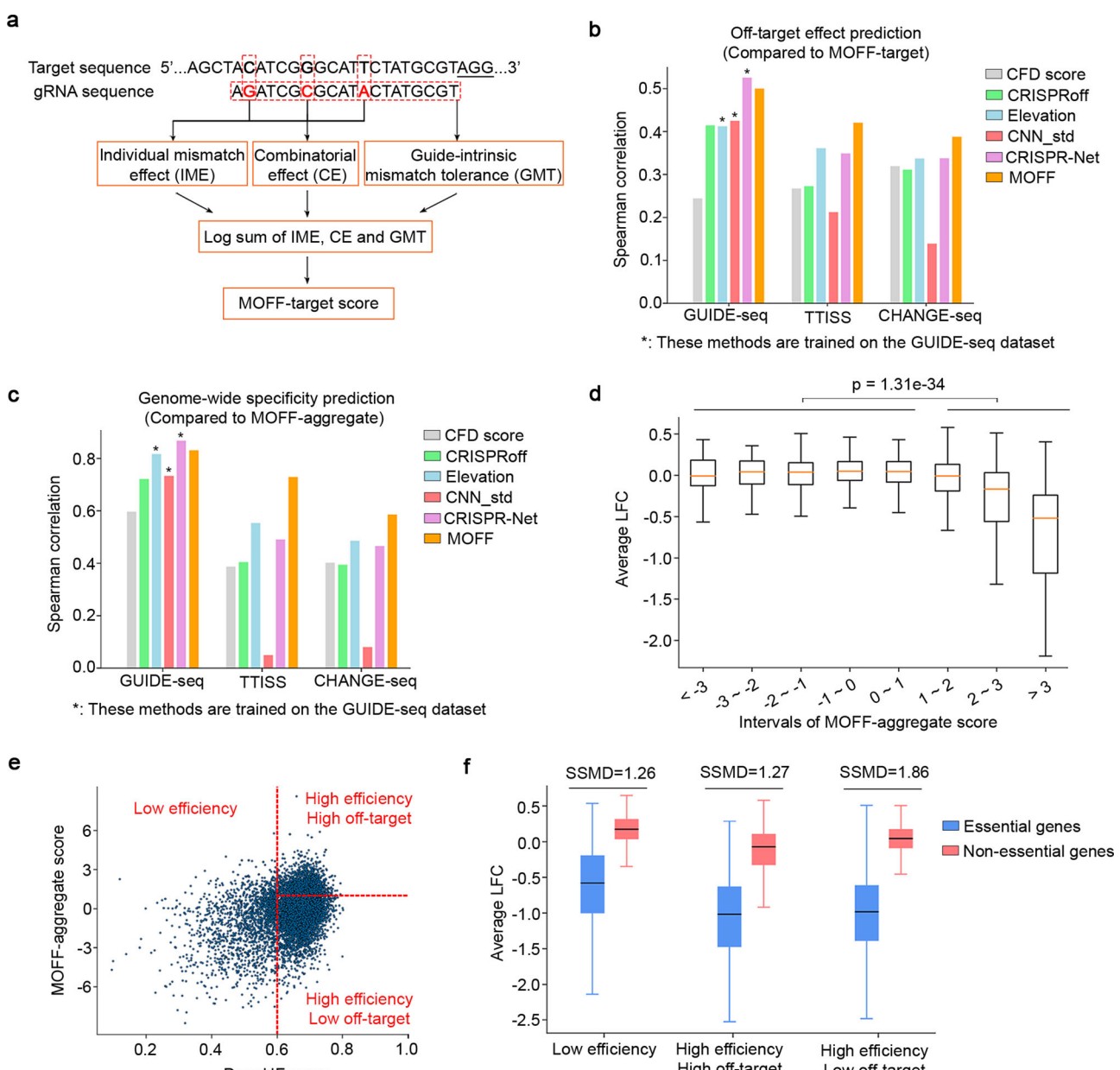

**Fig. 4 Prediction of off-target effect and gRNA specificity with MOFF. a** A schematic representation of the workflow of MOFF-target. **b**, **c** Comparison of the performance of **b** MOFF-target and **c** MOFF-aggregate to other methods in the prediction of **b** off-target effect at a specific genomic locus, or **c** genome-wide gRNA specificity defined as the total read counts at the off-target sites relative to the on-target read counts. The performance is measured by the Spearman correlation between the predicted and observed values at a log scale. See Methods section for details of method comparison. *: methods trained on the GUIDE-seq dataset. **d** 2408 gRNAs targeting non-essential genes[70] in the Avana library were grouped based on MOFF-aggregate scores. The box plot shows the distributions of average gRNA depletion (log-fold change, LFC) in viability screens across 342 cancer cell lines with respect to each group of gRNAs, $n = 59, 103, 234, 484, 686, 543, 209$, and 90 gRNAs within each score interval from left to right. The viability screening data were retrieved from the web portal of DepMap project[43]. The *p*-value was calculated using the two-tailed Mann–Whitney *U* test comparing the LFC of gRNAs with MOFF-aggregate scores smaller and larger than 1. The box plot displays a median line, interquartile range boxes, and min to max whiskers. **e** A scatter plot showing the categorization of 8021 gRNAs targeting core essential genes or non-essential genes in the Avana library, based on gRNA activity (DeepHF score[44], *x*-axis) and gRNA specificity (MOFF-aggregate score, *y*-axis). The red dashed lines used for gRNA categorization represent MOFF-aggregate score = 1 (horizontal) and DeepHF score = 0.6 (vertical), respectively. **f** A box plot comparing the distributions of the depletion of gRNAs targeting essential or non-essential genes in the DepMap viability screens. The data represent $n = 969$ essential, and 665 non-essential gRNAs in "low efficiency" group, $n = 945$ essential, and 842 non-essential gRNAs in "high efficiency high off-target" group, and $n = 3038$ essential, and 1566 non-essential gRNAs in "high efficiency low off-target" group. Strictly standardized mean difference (SSMD) scores are computed as the measure of effect size in the screens. The box plot displays a median line, interquartile range boxes, and min to max whiskers. Source data for Fig. 4b–f are provided in the Source Data file.

overfitting. Alternatively, MOFF-target and MOFF-aggregate, which are based on explicit sequence-dependent rules, achieved consistently superior performance in all three independent datasets. Furthermore, feature importance analyses suggest higher weights of IME and CE in MOFF-target. In contrast, the GMT effect significantly contributes to genome-wide specificity in MOFF-aggregate because it impacts the cleavage rates at all the off-target sites regardless of various contexts of mismatches (Supplementary Fig. 11).

Previous studies suggested that unintended off-target cleavages lead to decreased cell proliferation through induction of a $G_2$ cell cycle arrest[41,42]. To test if MOFF could select gRNAs to minimize the confounding off-target effect in the high-throughput CRISPR/Cas9 screens, we analyzed 7,403 gRNAs targeting 758 non-essential genes in the Avana and GeCKO-v2 libraries used in the large-scale DepMap project[43]. As expected, the MOFF-aggregate score is correlated with a cell depletion phenotype in viability screens (Fig. 4d and Supplementary Fig. 12). Further analysis showed that the combination of MOFF-aggregate and DeepHF[44], a gRNA efficiency prediction tool, optimizes gRNA selection to achieve greater effect size in CRISPR/Cas9 screens (Fig. 4e, f and Supplementary Fig. 13). Moreover, the gRNA selection based on MOFF and DeepHF leads to improved cross-library reproducibility on cell-specific essential genes (Supplementary Fig. 14). Collectively, these lines of evidence support the future application of MOFF for the rational design of customized CRISPR/Cas9 libraries.

### Improving allele-specific genome editing with mismatched gRNA.
Cas9-mediated allele-specific genome editing holds great potential for functional elucidation of disease-associated heterozygous mutations[45–49]. However, the selectivity of allele-specific targeting remains a significant challenge due to the sequence similarity between the mutant and wild-type alleles, which differ by only 1-bp for most point mutations[46,48,50,51]. The existing strategies are limited to the mutations that occur in the PAM or the seed regions[45,52]. Truncated 17-19 nt crRNAs improved the selectivity to a moderate degree but failed in some applications[46,48,50].

Given the observation of the "epistasis-like" combinatorial effect of two mismatches, we hypothesized that the selectivity of allele-specific targeting could be improved using a mismatched gRNA that differs by 1-nt from the mutant and 2-nt from the wildtype allele (Fig. 5a). To test this hypothesis, we used the dual-target synthetic system to perform a high-throughput allele-editing screen on 18 cancer hotspot mutations (Fig. 5b and Supplementary Data 5). In library design, the mutant and the wildtype sequences are encompassed in the dual-target site, and the gRNAs are of variable contexts corresponding to perfect match, truncated forms, and all possible single mismatches relative to the mutant. Among 2445 screened gRNAs, 574 (23.5%) showed high mutant-editing efficiency (>75% indel frequency at the mutant allele relative to perfectly matched gRNA) and high selectivity (<20% indel frequency at the wildtype allele relative to perfectly matched gRNA). 16 of the 18 cancer hotspot mutations are selectively targetable by mismatched gRNA, with the exception of 2 mutations lacking efficient mutant-editing potential. Using the same criteria, 4 and 9 cancer hotspot mutations are targetable by gRNAs of perfect match and in a truncated form, respectively (Table 1). The improvement made by mismatched gRNA was confirmed by experimental validations based on 6 selected cancer hotspot mutations, as shown in Fig. 5c. Importantly, the mismatched gRNAs achieve satisfactory selectivity for the mutations that occur in both seed and non-seed regions of the corresponding protospacers.

To test if the effective mismatched gRNAs can be predicted computationally, we applied MOFF-target to determine a subset of 364 mismatched gRNAs that are expected to be efficient at the mutant sequences and selective against the wildtype sequences (see Methods section for details). We found that the predicted gRNAs maintained a high level of mutant-editing efficiency comparable to the perfectly matched gRNAs, and significantly improved the selectivity when compared to the truncated 17-18 nt forms (Fig. 5d, e). Among the predicted mismatched gRNAs, 191 (52.5%) are effective in the high-throughput screen, confirming the feasibility of allele-specific editing using the computer-aided design of mismatched gRNAs. To address the needs of guide design in allele-specific editing, we implemented a module in the MOFF package, named MOFF-allele, that allows the users to select the optimal mismatched gRNA for their study.

### Discussion
In this study, we devised a high-throughput dual-target synthetic system to explore the sequence features associated with CRISPR/Cas9 off-target effect. We determined the rules underlying a GMT and the combinatorial effect of multiple mismatches. The sequence-dependent rules are significantly associated with the free-energy landscape during R-loop formation and support a recent multi-state kinetic model[40]. Of note, R-loop formation is not only required for the DNA cleavage mediated by Cas9 and its variants, but also for other CRISPR techniques based on dCas9 and Cas12a[53,54]. Therefore, we anticipate that these sequence-dependent rules and the kinetic model are generally applicable to a wide spectrum of CRISPR applications such as dCas9-mediated transcriptional repression and activation, base editing, and prime editing[55–60], at least to a significant extent. In addition to R-loop formation, the interaction between Cas9 and unwound target DNA strand, the conformational change of Cas9-gRNA structure, as well as the folding stability of gRNAs, collectively contribute to the sequence-dependent free-energy landscape. Therefore, the kinetic model can be further improved by taking these factors into consideration, which will lead to a better explanation of off-target effect from a biophysical perspective.

A simple combination of our rule sets led to the development of MOFF, a model-based off-target prediction method that outperforms previous methods including complex deep learning frameworks. We ascribe the improvement to two factors. First, the previous methods were developed on datasets with a limited number of gRNAs that are insufficient for modeling on a large feature space spanned by 20-nt nucleotides. Second, complex models are sensitive to platform-specific biases and are prone to overfitting, whereas rule-based approaches such as MOFF are more robust against the variation of platforms. The second point is supported by our additional evaluation results as shown in Supplementary Fig. 15, in which MOFF outperforms traditional machine learning methods (Gradient Boosted Tree, Random Forest Regressor, and Support Vector Machine) when trained on our dual-target dataset and tested on public datasets. Meanwhile, we acknowledge that complex machine learning models are advantageous in extracting subtle features that are indiscernible by simple rule mining if the platform-specific biases are minimal. We envision that MOFF can be further improved by: (i) supplying advanced machine learning techniques with known rules and biophysical laws, followed by training the model on a compendium of independent datasets; (ii) considering DNA/RNA bulges, which account for ~10% of genomic off-targeting events[61]; (iii) accounting the mismatches in the PAM sequence, as some alternative PAM sequences, such as NAG, also lead to active Cas9/gRNA editing[9]; and (iv) incorporating sequence-

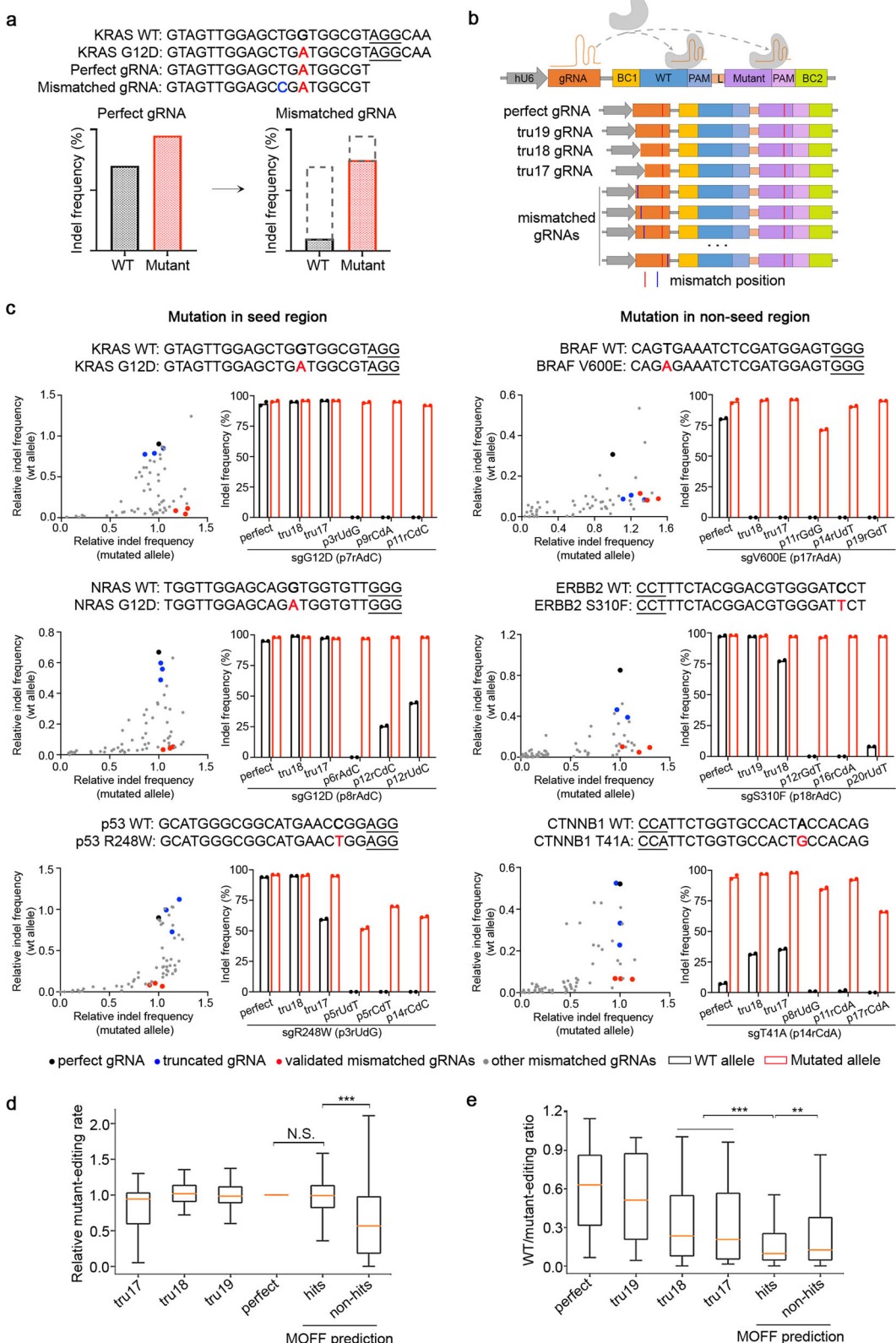

independent features such as chromatin structure and epigenetic markers, as reported previously[19,62].

Current strategies for allele-specific editing using discriminating gRNAs are mainly focused on the mutations in the PAM or the seed region. Although several successful applications have been reported using Cas9 and its orthologs/variants[45,46,49], the selectivity of allele-specific editing remains a significant challenge.

Based on the "epistasis-like" combinatorial effect of dual mismatches that we reported, we propose to use mismatched gRNAs to improve the selectivity of allele-specific genome editing, where the intended allele harbors a single tolerable mismatch relative to the gRNA and the unintended allele harbors two intolerable mismatches. The high-throughput allele-editing screen confirmed that a vast majority of cancer hotspot mutations located either in

**Fig. 5 Improvement of allele-specific genome editing with mismatched gRNA. a** A conceptual illustration of improving the selectivity of allele-specific editing using mismatched gRNA exemplified by KRAS G12D sequence. Nucleotides highlighted in colors represent the original mutation (red) and the introduced mismatch (blue). PAM sequence was indicated by an underline. **b** A schematic representation of experimental design of the allele-editing library using the dual-target system. **c** Experimental validations of the selectivity of allele-specific editing on 6 selected cancer hotspot mutations. The scatter plot of each mutation on the left shows the result of the high-throughput allele-editing screen. Relative indel frequency to wildtype or mutant allele was normalized by the indel frequency of perfect gRNA on the mutant allele. The bar chart on the right displays the validation results of indel frequency on wildtype and mutant alleles using one perfect, two truncated, and three effective mismatched gRNAs in HEK293T cells integrated with wildtype or mutant allele-target sequence. There are 2 biological replicates for each genomic sample. **d** Comparison of relative mutant-editing rate on MOFF predicted hits to perfect gRNA and MOFF predicted non-hits. **e** Comparison of the selectivity using wildtype to mutant-editing ratio on MOFF predicted hits to 17-18 nt truncated gRNAs and MOFF predicted non-hits. **d, e** $**p < 0.01$, $***p < 0.001$, N.S.: not significant. The data represent $n = 29$ (tru17), 32 (tru18), 35 (tru19), 35 (perfect), 349 (hits), and 1,659 (non-hits) gRNA-targets pairs in the screen. The box plots display a median line, interquartile range boxes and min to max whiskers. The exact $p$-values from left to right are: **d** $p = 0.55$, $p = 4.78e-41$, and **e** $p = 1.88e-04$, $p = 5.02e-03$. The $p$-values were calculated using two-tailed Manny–Whitney $U$ test. Source data for Fig. 5c–e are provided in the Source Data file.

**Table 1 Summary of the selectivity of allele-specific editing on 18 cancer hotspot mutations using perfect, truncated, and mismatched gRNAs.**

| Gene | Protein change | Nucleotide change | Targetability | | |
|---|---|---|---|---|---|
| | | | Perfect gRNA | Truncated gRNA | Mismatched gRNA |
| *ERBB2* | L755S | c.2264T>C | Yes | Yes | Yes |
| *IDH1* | R132H | c.395G>A | Yes | Yes | Yes |
| *PIK3CA* | H1047R | c.3140A>G | Yes | Yes | Yes |
| *VHL* | R200W | c.598C>T | Yes | Yes | Yes |
| *BRAF* | V600E | c.1799T>A | No | Yes | Yes |
| *EGFR* | L858R | c.2573T>G | No | Yes | Yes |
| *EGFR* | T790M | c.2369C>T | No | Yes | Yes |
| *GNAS* | R201C | c.601C>T | No | Yes | Yes |
| *KRAS* | Q61H | c.183A>C | No | Yes | Yes |
| *CTNNB1* | T41A | c.121A>G | No | No | Yes |
| *ERBB2* | S310F | c.929C>T | No | No | Yes |
| *FBXW7* | R465C | c.1393C>T | No | No | Yes |
| *FGFR3* | Y373C | c.1118A>G | No | No | Yes |
| *KRAS* | G12D | c.35G>A | No | No | Yes |
| *NRAS* | G12D | c.35G>A | No | No | Yes |
| *TP53* | R248W | c.742C>T | No | No | Yes |
| *PIK3CA* | E542K | c.1624G>A | No | No | No[*] |
| *RAC1* | P29S | c.85C>T | No | No | No[*] |

[*]Mutants lacking efficient gRNA for targeting.

the seed region or non-seed region are selectively targetable through the computer-aided design of mismatched gRNAs. This improvement expands the application domain of allele-specific editing as a time-saving and cost-effective approach for the perturbation of endogenous mutant alleles, thus benefiting functional studies of disease- or trait-associated heterozygous point mutations in a variety of species. Besides allele-specific editing, mismatched gRNAs can be applied to selectively edit one of the paralogous genes that share high degrees of sequence homology, which will facilitate the elucidation of genetic interactions and functional crosstalk between paralogs. Moreover, while our validations are focused on cancer hotspot mutations, the strategy of computer-aided design of mismatched gRNAs holds the clinical potential to correct dominant-negative mutations that drive mendelian disorders, which will be addressed in future work.

## Methods

**Oligonucleotide library design**. In this study, we designed 11 high-throughput libraries for systematically decomposing sequence determinants that govern CRISPR/Cas9 specificity. Each library consists of ~12,000 oligonucleotides. These 11 libraries are summarized as follows and more details are provided in Supplementary Data 6.

Library T1, T2, and T3: paired gRNA and dual-target design (Fig. 1b) for method evaluation and rule mining. The T1 library includes control gRNAs, benchmark gRNAs and random gRNAs for method evaluation. The T2 and T3

libraries, together with the fraction of random gRNAs in T1, are used to uncover the rules underlying off-targeting.

Library S1: paired gRNA and single-target design (Fig. 1a) for method evaluation. We designed this library following a previously described high-throughput library design protocol[34]. For a systematic comparison, Library S1 contains the same sets of gRNA and corresponding target combination as Library T1.

Library Allele: paired gRNA and dual-target design (Fig. 1b) for the allele-editing screen. We designed this allele-editing library to leverage the mismatched gRNAs to target 18 cancer hotspot mutations, which were curated from Memorial Sloan Kettering Cancer Center (MSKCC) Cancer Hotspots (https://www.cancerhotspots.org) and cBioportal for Cancer Genomics (https://www.cbioportal.org) with reported gain-of-function or dominant-negative effects. For dual-target design, the wildtype sequence is placed at the off-target position and the mutant sequence at the on-target position (Fig. 5b).

Library sg1, sg2, sg3, sg4, sg5, and sg6: target-only design (no gRNA pooling) for GMT exploration (Fig. 2a, b). Two gRNAs, sg2 and sg6, have the crystal structure with Cas9 and their targets (PDBs: 4OO8 and 4UN3). Target sequences in each library include the on-target, all the possible 1 mismatch (1-MM, 66 sequences), 2 mismatches (2-MM, 2,079 sequences) targets, and randomly selected 3 mismatches (3-MM, 100 sequences), 4 mismatches (4-MM, 100 sequences), 5 mismatches (5-MM, 100 sequences), and 6 mismatches (6-MM, 47 sequences) targets.

The gRNAs in these 11 libraries are categorized into 5 types: control gRNAs, benchmark gRNAs, randomly generated gRNAs, allele-editing gRNAs, and truncated gRNAs. Every control gRNA was designed with 4 types of dual-target sequences by the replacement of "NGG" with "NTT" sequence at the PAM to represent 4 combinations of cleavage events (no cleavage, left, right, and both) (Fig. 1c). The benchmark gRNAs with reported genomic off-target sites were

curated from different methods detecting genome-wide off-targets induced by Cas9 editing in vitro or in vivo and used to evaluate the dual-target approach[1,12,17,36]. Unreported genomic off-target sites with <5 mismatches relative to corresponding benchmark gRNAs were generated by CRISPRitz[63] (Fig. 1h–j). Except for the gRNAs in Libraries sg1-6, each randomly generated gRNA was designed with 7 types of off-target sequences to systematically decompose the estimated off-on ratios into single-mismatch effect, combinatorial effect, and GMT. These seven sequence types include three targets with 1-MM, 3 with 2-MM, and 1 with 3-MM. The mismatches in the 2-MM and 3-MM targets are the combinations of the mismatches in 1-MM targets (Fig. 1k). Frequencies of four nucleotide types (A, G, C, and T) at each position for randomly designed gRNAs are roughly evenly distributed (Supplementary Fig. 16). Allele-editing gRNAs were designed to test the selectivity for allele-specific targeting on 18 cancer hotspot mutations (Fig. 5b). Truncated 17–19 nt gRNAs were designed to compare to mismatched gRNAs for allele-editing assessment (Fig. 5d, e).

In the paired gRNA and dual-target libraries (Library T1, T2, T3, and Allele), each 190-bp oligonucleotide consists of an 18-bp left primer sequence, a 10-bp barcode 1, a 23-bp corresponding off-target sequence containing a PAM, a 15-bp linker to segregate two targets, a 23-bp corresponding on-target sequence containing a PAM, a 15-bp barcode 2, a 45-bp cloning linker sequence containing a vector homology sequence, an SspI enzyme recognition site and an hU6 homology sequence, a 21-bp guide sequence beginning with "g" ($gN_{20}$), and a 20-bp right primer sequence. In the paired gRNA and single-target library (Library S1), each 157-bp oligonucleotide consists of a longer 20-bp barcode 2 but lacks the 15-bp linker and the second 23-bp target. All other sequence components are identical between the single- and dual-target design. In the target-only libraries (Library sg1, sg2, sg3, sg4, sg5, and sg6), each 109-bp oligonucleotide consists of a 25-bp left primer sequence, a 10-bp barcode 1, a 27-bp corresponding target sequence containing a PAM, a 20-bp barcode 2, and a 27-bp right primer sequence (Supplementary Note). All the oligonucleotides in these libraries passed the filters of having no thymine homopolymers more than three nucleotides long in gRNA sequences, carrying no more than one SspI enzyme recognition site, one left primer site, and one right primer site, and no sites besides the target sequence being cut by gRNAs in the constructed plasmids.

**Plasmid cloning**. gRNA I and D associated with distinct repair mechanisms upon double-strand breaks were selected from a previous single-target library predicting mutations generated by CRISPR/Cas9. Dual-target I and D were designed with four types of dual-targets by the replacement of "NGG" with "NTT" sequence at the PAM as described above. For gRNA I/D and corresponding dual-target I/D pairs cloning, the 152-bp single-strand Ultramer™ DNA oligonucleotides with 4 types of dual-target I and D were synthesized (IDT) and amplified with Q5 High-Fidelity DNA polymerase (NEB #M0492). Meanwhile, gRNA I and D oligonucleotides (Sigma) were annealed and ligated to the BsmBI (NEB #R0739)-linearized LentiGuide-BSD-PacI plasmid. The LentiGuide-BSD-PacI plasmid was derived from the LentiGuide-BSD plasmid by inserting a PacI recognition site to 33-bp downstream of the scaffold sequence using Q5 site-directed mutagenesis (NEB #E0552). The resulting plasmids were then linearized by PacI (NEB #R0547), and undergone Gibson assembly (NEBuilder HiFi DNA Assembly Master Mix, NEB #E2621) with the amplified four types of dual-target I and D fragments. Assembled products were then transformed to homemade Stbl3 competent cells, and the plasmids were extracted using QIAprep Spin Miniprep Kit (QIAGEN) and confirmed by Sanger sequencing.

For plasmid library cloning, the oligonucleotide libraries with paired gRNA and dual-target design were synthesized by Twist Bioscience, and libraries with paired gRNA and single-target design or target-only design were ordered from CustomArray. The plasmid library containing pooling gRNAs and corresponding single- or dual-target pairs was prepared by undergoing an intermediate circulation, SspI enzyme linearization, and Gibson assembly into a lentiviral plasmid LentiGuide-BSD-Δscaffold. This multistep cloning procedure was adapted and modified from previously described protocols[33,34]. Briefly, the oligonucleotide pools were amplified with Q5 High-Fidelity DNA polymerase for extension of the oligonucleotide sequences with overhangs complementary to a donor G-block. Gibson assembly was performed to ligate the amplified pools to the 125-nt donor gBlock fragment (IDT) encoding the gRNA scaffold at a molar ratio of 1:3 at 50 °C for 1 h, and the resulting products were incubated with Plasmid Safe ATP-Dependent DNase (Lucigen #E3101K) at 37 °C for 1 h to remove the linear fragments. The assembled circular DNA was purified with QIAquick PCR Purification Kit (QIAGEN), followed by linearization with SspI enzyme (NEB #R3132) at 37 °C for 4 h, and purification with QIAquick PCR Purification Kit. Next, the purified products were performed the second amplification with Q5 High-Fidelity DNA polymerase for the addition of overhangs complementary to gRNA-expressing plasmid LentiGuide-BSD-Δscaffold, and the PCR products were then purified with QIAquick Gel Extraction Kit (QIAGEN). This LentiGuide-BSD-Δscaffold plasmid lacking the scaffold sequence was generated by removing the gRNA scaffold from LentiGuide-BSD plasmid using Q5 site-directed mutagenesis. Gibson Assembly was then employed to fuse the second amplified oligonucleotide pools to BsmBI-linearized LentiGuide-BSD-Δscaffold at a molar ratio of 3:1 at 50 °C for 1 h. The resulting products were then purified by isopropanol precipitation with GlycoBlue™ Coprecipitant (Thermo Fisher Scientific

#AM9516), dissolved in TE buffer (pH 8.0) to a final concentration of 100 ng/μl at 55 °C for 10 min, and transformed into Endura electrocompetent cells (Lucigen #60242). The plasmid library was extracted by NucleoBond Xtra Maxi Kit (MACHEREY-NAGEL). The construction of the plasmid library containing single gRNA and corresponding targets followed the cloning processes of gRNA I/D and dual-target I/D pairs as mentioned above. In brief, oligonucleotides of each gRNA were annealed and inserted into BsmBI-linearized LentiGuide-BSD-PacI, and the constructs then were verified by Sanger sequencing. In the meantime, the target-only library was amplified with Q5 High-Fidelity DNA polymerase and purified with gel purification. Gibson Assembly was applied to ligate the amplified target-only oligonucleotide pools to their corresponding PacI-linearized single gRNA-expressing plasmids LentiGuide-BSD-PacI at a molar ratio of 6.5:1 at 50 °C for 1 h. The assembled products were purified by isopropanol precipitation and transformed into electrocompetent cells, followed by plasmid extraction as described above.

To validate the results of the high-throughput allele-editing screen, 6 genes with cancer hotspot mutations were selected. For allele-editing validation plasmid cloning, oligonucleotides containing cancer hotspot mutation-derived wildtype and mutant target sites, also referred to as wildtype and mutant target, were synthesized (IDT) and annealed, followed by insertion into BsmBI-linearized plasmid LentiEGFP-P2A-Puro. Meanwhile, 6 gRNAs were designed to target each selected cancer hotspot mutation, including one perfectly matched, two truncated, and three effective mismatched gRNAs based on the screen results. LentiCRISPR-V2-BSD (Addgene plasmid #83480) was employed to insert each gRNA according to the protocol from Feng Zhang's lab. The ligated products were extracted using QIAprep Spin Miniprep Kit and confirmed by Sanger sequencing.

All the primers, gRNA, and fragments used for plasmid construction are listed in Supplementary Data 7.

**Cell culture**. HEK293T cell line was purchased from ATCC (CRL-3216) and cultured in DMEM medium (Gibco) supplemented with 10% FBS (Sigma) and 1% penicillin-streptomycin (Gibco) at 37 °C with 5% $CO_2$. A monoclonal HEK293T cell line expressing doxycycline-inducible Cas9 (HEK293T-iCas9) was generated by infecting parental cells with pCW-Cas9 (Addgene plasmid #50661) lentivirus. All the cell lines are routinely tested for being free of mycoplasma contamination using MycoAlert™ Mycoplasma Detection Kit (Lonza #LT07-218).

**Lentivirus production**. HEK293T cells ($5 \times 10^6$) were seeded in a 10-cm tissue culture dish 1 day before transfection. On the day of transfection, 4 μg of lentiviral plasmid, 4 μg of psPAX2 (Addgene plasmid #12260), and 2 μg of pMD2.G (Addgene plasmid #12259) in 500 μl Opti-MEM (Gibco) were mixed with 25 μl X-tremeGene HP DNA transfection reagent (Roche #06366236001) and then added to the pre-seeded HEK293T cells. The culture medium was changed with the addition of 20 μM HEPES buffer 6 h after transfection. Lentiviral supernatant was harvested and filtered through a 0.45-μM syringe filter (Millipore) 48 h after transfection. Aliquots were frozen at −80 °C for later use.

**CRISPR screen and deep sequencing**. HEK293T-iCas9 cells were seeded in eighteen 10 cm tissue culture dishes ($4 \times 10^6$ cells per dish) with three independent biological replicates 1 day before transduction. On the day of transduction, the lentiviral plasmid library was transduced into the cells at an MOI of 0.3 in the presence of 8 μg/ml polybrene (Millipore). The infected cells were passaged to 15 cm tissue culture dishes 48 h after infection, selected with 20 μg/ml blasticidin (InvivoGen #ant-bl-1) for 3 days, and then incubated with doxycycline (Sigma #D9891) to induce Cas9 expression. Cells were passed every 2–3 days with at least $2 \times 10^7$ cells for each passage to maintain the library diversity and harvested ($2 \times 10^7$ cells per replicate) on days 0, 3, 5, and 9 after Cas9 induction. Cell pellets were frozen at −80 °C for later genomic extraction using Blood & Cell Culture DNA Midi Kit (QIAGEN).

For the transduction of lentiviral plasmids with gRNA I/D and corresponding dual-target I/D, HEK293T-iCas9 cells were seeded in six-well plates ($5 \times 10^4$ cells per well) with two biological replicates following the same procedure as described above. In all, $2 \times 10^6$ cells were harvested on day 9 after Cas9 induction for genomic extraction using QIAamp® DNA Mini Kit (QIAGEN).

After genomic extraction, the integrated target sequences were PCR amplified using NEBNext® Ultra™ II Q5® Master Mix (NEB #M0544) and prepared for deep sequencing by 2 rounds of PCR as described previously[64]. The 1st round PCR (16 cycles) was performed with 40 μg genomic DNA divided into eight 100 μl-PCR reactions to achieve 500-fold coverage over the library. To attach the Illumina adaptor and barcode sequences, the 2nd round PCR (12 cycles) was conducted with 5 μl of the 1st purified PCR product. The gel-purified samples were pooled and sequenced on Illumina Miseq, Hiseq 3000 or Nextseq 500 by 75-bp paired-end sequencing at MDACC-Smithville Next Generation Sequencing Core. All the primers applied for sequencing library preparation are provided in Supplementary Data 7.

**Allele-specific editing validation**. To validate the allele-editing results from the high-throughput screen, 2 rounds of transduction were performed. HEK293T cells were seeded in six-well plates ($1 \times 10^5$ cells per well) and infected with lentivirus of

LentiEGFP-P2A-Puro harboring the wildtype or mutant target sequence related to the 6 selected cancer hotspot mutations. On day 4 after puromycin (InvivoGen #ant-pr-1) selection, the resulting HEK293T cells were re-seeded in six-well plates ($1 \times 10^5$ cells per well), followed by infection with lentivirus of LentiCRISPR-V2-BSD expressing perfectly matched, truncated or mismatched gRNAs. The infected cells from 2 biological replicates were passaged every 2–3 days and harvested ($2 \times 10^6$ cells) on day 9 after blasticidin selection for genomic extraction.

Genomic DNA was isolated from cell pellets with QIAamp® DNA Mini Kit (QIAGEN) according to the manufacturer's protocol. In total, 2 µg genomic DNA of each sample was used to amplify the integrated wildtype or mutant target sequence with Q5 High-Fidelity DNA polymerase. PCR products were purified by gel purification and undergone Sanger sequencing at MDACC-Smithville Molecular Biology Core. The wildtype and mutant allele-editing frequencies by perfectly matched, truncated, and mismatched gRNAs were analyzed through a Sanger sequencing-based CRISPR analysis online tool Inference of CRISPR Edits (ICE, https://ice.synthego.com).

**Sequencing data analysis**. A computational pipeline was developed to process the generated sequencing data. We first combined the partially overlapped, paired-end reads into a single sequence using FLASH v1.2.11[65] with the options "-m 20, -M 150, -x 0.1, -allow-outies" (a minimum overlap of 20 bp, a maximum overlap of 150 bp, a maximum mismatch density of 0.1 and allow outie pairs). Next, we extracted the unique barcode 1 and 2 pairs from the merged reads and compared them to the barcodes of the designed constructs in the library. Sequencing reads were assigned to a specific construct if their barcodes were perfectly matched to the 10-nt of barcode 1 and last 10 bp of 15-nt of barcode 2 of the construct. The sequences between the two barcodes were then aligned to the designed sequences for indel calling using the Smith-Waterman algorithm[66] with gap open penalty of 8, mismatch penalty of 1, match score of 2, and mismatch score of −2. Finally, the sequencing reads were assigned to one of five different indel types, i.e., wt + wt, indel + wt, wt + indel, indel + indel and large deletion, based on the alignment results. Specifically, if a deletion larger than 32-bp was identified in the alignments, the read was assigned to the large deletion category. Otherwise, we adopted a 10-bp window around the cutting site to capture the potential CRISPR-induced editing at on-target and off-target sequence. We can then easily assign the reads into four other categories: indel + wt (indels detected at on-target site only), wt + indel (indels detected at off-target site only), indel + indel (indels detected at both on- and off-target sites), and wt + wt (indels detected at neither on- nor off-target sites). Of note, we only observed rare cases (~0.005%) where two cleavage sites are inverted in our dual-target design[67]. This is consistent with previous report where inversion rate of small DNA fragments with size <100 bp is much lower compared to that of larger fragments of hundreds to thousands bps[68]. Thus, here we did not include this inversion type in the classification of indel types. Distributions of the number of sequenced reads per gRNA-target pair to indicate the sequencing depths of the experiments with dual-target design are shown in Supplementary Fig. 17.

**Estimation of off-on ratios**. We developed a statistical method to infer the read counts of different cleavage states from the observed editing types of the sequencing data. Given a gRNA-target pair, we denote $C = [c_1, c_2, c_3, c_4, c_5]^T$ to represent the observations of read counts, where $c_1$ to $c_5$ correspond to the editing types of wt + wt, indel + wt, wt + indel, indel + indel, and large deletion, respectively. We denote $S = [s_1, s_2, s_3, s_4]^T$ to represent the numbers of reads in the four cleavage states, where $s_1$ to $s_4$ correspond to the states of "no cleavage", "cleavage at off-target only", "cleavage at on-target only", and "both cleavage", respectively. We model the observation $C$ to be:

$$C = P^T S + \varepsilon \qquad (1)$$

where $P$ is a $4 \times 5$ stochastic matrix and $\varepsilon$ is a vector of Gaussian noise. An entry $p_{i,j}$ ($i = 1, 2, 3, 4$; $j = 1, 2, \ldots, 5$) in the matrix $P$ represents the conditional probability of observing the $j$th editing type given the $i$th cleavage state. The sum of each row in $P$ is 1, i.e., $\sum_{j=1}^{5} p_{i,j} = 1, \forall i$. Given the model in Fig. 1e, $P$ can be written as:

$$P = \begin{bmatrix} p_{1,1} & p_{1,2} & p_{1,3} & p_{1,4} & p_{1,5} \\ 0 & p_{2,2} & 0 & 0 & 0 \\ 0 & 0 & p_{3,3} & 0 & p_{3,5} \\ 0 & 0 & 0 & p_{4,4} & p_{4,5} \end{bmatrix} \qquad (2)$$

The parameters in $P$ were estimated from the observations of read counts on the control dual-target sequences. The first row, denoted $P_1$, represents the background noise and PCR artifacts, which were estimated based on the read distribution of the NTT + NTT (no cleavage) controls. The second row, denoted $P_2$, was estimated based on the observations from the NGG + NTT controls. Depending on the gRNA cutting efficiency, the reads derived from the NGG + NTT controls are subject to a mixture of the "no cleavage" and "cleavage at off-target only" states. Given the estimated $P_1$ and the read counts of the five editing types on the NGG + NTT controls, $P_2$ can be explicitly estimated by solving linear equations. Similarly, the parameters in the third and fourth rows in $P$ were computed from the observations on the NTT + NGG and NGG + NGG controls.

Given the stochastic matrix $P$, we adopted a maximum-likelihood estimation (MLE) approach to compute $S$. This can be achieved by minimizing $\|P^T S - C\|_2^2$, under the constraint of $s_i \geq 0, \forall i$. With the estimated reads for the four cleavage states, the off-on ratio of a gRNA-target pair was calculated as:

$$r = \frac{s_2 + s_4}{s_3 + s_4} \qquad (3)$$

Where $s_2$, $s_3$ and $s_4$ are the estimated read counts for "cleavage at off-target only", "cleavage at on-target only", and "both cleavage", respectively. Considering that the calculation of off-on ratios is associated with large variation using a limited number of reads in the denominator, we filtered out gRNA-target pairs with less than 100 total edited reads at on-target site, i.e., $s_3 + s_4 < 100$, resulting in a total number of 10,460 gRNA-target pairs with the estimated off-on ratios (Supplementary Data 1). When computing log off-on ratio, we added a small constant $c$ to control the variation, i.e., $\log(r + c)$. The constant $c$ is chosen to be 0.01 because the dynamic range of off-on ratio evaluation is approximately 0.02–1.

**Estimation of GMT**. We assume that the intrinsic sequence of the gRNA and the mismatch context between gRNA and target sequence jointly contribute to the overall off-target effects. To decompose these two factors, we collected the off-on ratios of 3897 1-mismatch targets corresponding to 1438 gRNAs in our dataset (Supplementary Data 4). There are 12 types of unmatched nucleotide pairs (A-C, A-G, …, U-C), and 20 different positions of the mismatches, resulting in 240 mismatch types in total. Suppose $r_{ij}$ represents the off-on ratio of the $i$th gRNA with a mismatch of the $j$th type, we model $r_{ij}$ to be the multiplication of mismatch-dependent effect $m_j$ and gRNA-intrinsic mismatch tolerance $g_i$. In a log scale, this can be written as:

$$\text{Log}(r_{ij} + c) = \log(m_j) + \log(g_i) + \varepsilon \qquad (4)$$

where $\varepsilon$ is the Gaussian noise and $c$ is the small constant for controlling the variation of log off-on ratio. We then applied a gradient descent algorithm to compute the MLE estimation of $m_j$ and $g_i$, which represents the mismatch-dependent effect and the gRNA-intrinsic mismatch tolerance, respectively. The mismatch-dependent effects for single-mismatches are presented as a $12 \times 20$ matrix, namely M1 matrix, in which each entry represents the effects of a nucleotide mismatch at a specific position of the target sequence. The estimated $g_j$ was subsequently used to explore the sequence determinants of GMT.

**Prediction of GMT**. To predict GMT from a gRNA sequence, we took all the 1438 gRNAs with estimated GMT scores ($g_j$) to train a CNN model. For each gRNA, we tested two different encoding strategies, i.e., mononucleotide and dinucleotide, to vectorize the gRNA sequence as inputs (Supplementary Fig. 8). For the mono-nucleotide encoding, a 20-nt gRNA sequence was binarized into a $4 \times 20$ two-dimension array, with 0 s and 1 s indicating the absence or presence of 4 different nucleotides (A, T, C, G) at every single position. For the dinucleotide encoding, 20-nt gRNA sequence was binarized into a $16 \times 19$ two-dimension array, with 0 s and 1 s indicating the absence or presence of 16 different dinucleotides (AT, AC, AG, AA, TT, TA, TG, TC, CC, CA, CG, CT, GG, GA, GT, GC) at each of the constitutive positions along the gRNA sequence. A CNN regression model was then designed using Keras (https://keras.io/) with a TensorFlow backend engine, consisting of one convolution layer and one dense layer, terminating in a single neuron.

We compared two encoding methods for data vectorization with different settings of parameters including the size, shape, and number of convolution kernels using a five-fold cross-validation strategy. The performance was assessed by computing Spearman's correlation between the predicted and observed GMT scores. Finally, we selected the dinucleotide inputs and CNN model with three kernels in the convolutional layer. We further tested our model on two independent datasets: the TTISS dataset, which includes 59 sgRNAs with genome-wide off-targets detected by TTISS for nine different Cas9 variants[18]; the CHANGE-seq dataset, which includes 110 gRNAs with genome-wide SpCas9 off-targets detected by CHANGE-seq[19]. For each gRNA, an overall off-on ratio was calculated as the sum of detected off-target reads divided by on-target reads. We predicted the GMT scores for all the gRNAs and compared the overall off-on ratios of gRNAs with high (top 25%) and low (bottom 25%) GMT scores. Statistical significance was measured using Mann–Whitney $U$ test.

**Estimation of combinatorial effects**. We denote $\delta_{ij}$ as the position-dependent combinatorial effect between two mismatches that occur at the $i$th and the $j$th nucleotide relative to the PAM ($i, j = 0, 1, 2, \ldots, 20$). A value of $\delta_{ij}$ close to 1 suggests "independence" of the combination, whereas $\delta_{ij}$ close to 0 suggests a strong "epistasis-like" combinatorial effect. We estimated $\delta_{ij}$ from the off-on ratios of 2-MM and individual 1-MM targets, as specified below.

To obtain a sufficient number of data points for a robust estimation of $\delta_{ij}$, we consider the mismatches located in the $i$th and the $j$th nucleotide, as well as those in the adjacent locations. Suppose there are $N_{ij}$ 2-MM targets with mismatches at positions (i, j), (i, j-1), (i, j+1), (i-1, j), and (i+1, j), we denote $x_{ij}^k$ to be the off-on

ratio of the $k$th 2-MM target in this group ($k = 0, 1, 2, \ldots, N_{ij}$) and model $x_{ij}^k$ as:

$$x_{ij}^k = y_{ij}^k z_{ij}^k \delta_{ij} + \varepsilon \tag{5}$$

where $y_{ij}^k$ and $z_{ij}^k$ are the off-on ratios of the 1-MM targets corresponding to the 2-MM target in the library design, and $\varepsilon$ is the Gaussian noise. The MLE estimate of $\delta_{ij}$ can be explicitly computed as:

$$\hat{\delta}_{ij} = \sum_{k=1}^{N_{ij}} \left( x_{ij}^k y_{ij}^k z_{ij}^k \right) \Big/ \sum_{k=1}^{N_{ij}} \left( y_{ij}^k z_{ij}^k \right)^2 \tag{6}$$

The combinatorial effects for 2-position combinations were presented as a $20 \times 20$ matrix, namely M2 matrix, in which each entry represents the combinatorial effects between two mismatched positions.

To cross-reference the combinatorial effects derived from our data, we also calculated the relative co-occurrence score (RCS) that represents the observed frequency of two mismatches relative to random expectation based on CHANGE-seq dataset[19]. Assuming that there are $n$ off-target sites detected in the genome, the RCS is defined as:

$$RCS = \frac{a_{ij} * n}{b_i * c_j} \tag{7}$$

Where $a_{ij}$ is the number of off-target sites harboring mismatches at both positions $i$ and $j$, $b_i$ is the number of target sites harboring mismatches at position $i$, and $c_j$ is the number of target sites harboring mismatches at position $j$. We computed the RCS for all the gRNAs in the CHANGE-seq dataset and took the average RSC over the gRNAs to obtain the matrix in Fig. 3e. To explore the combinatorial effect from a biophysical point of view, we used a previous kinetic model[40] to perform simulation and to estimate $\delta_{ij}$ from the simulated data. In brief, the cleavage rate can be directly calculated given the free energy gain of the binding the PAM, the energy gain for extending the hybrid over a match, the cost associated with extending the hybrid over an isolated mismatch, as well as the cost of extending the hybrid over neighboring mismatches. As an illustration, we plotted the combinatorial effect calculated for a specific choice of these parameters with the gain in energy due to PAM binding of 5 $k_B T$, the gain per correctly matched hybrid base pair of 0.25 $k_B T$, the cost of a mismatch of 4 $k_B T$ if it is isolated and 3 $k_B T$ if it is added to an existing bubble (Fig. 3g).

**Model-based off-target prediction with MOFF.** In MOFF, we integrate three factors including the individual mismatch effect (IME), the combinatorial effect (CE), and the GMT effect.

To explain the MOFF model, we start with a gRNA $g$ and a target with a single mismatch $m_1$. The expected off-on ratio, $S(g, m_1)$, is the multiply of IME and GMT, i.e.,

$$S(g, m_1) = s_1 s_{GMT} \tag{8}$$

where $s_1$ is the effect of $m_1$ computed from the M1 matrix as described in "Estimation of GMT" section, and $s_{GMT}$ is the GMT effect estimated from the dinucleotide CNN regression model as described in "Prediction of GMT" section.

Next, we consider two mismatches $m_1$ and $m_2$. The expected off-on ratio, $S(g, m_1, m_2)$, is the multiply of the effects of individual single mismatches and the combinatorial effect:

$$S(g, m_1, m_2) = S(g, m_1) S(g, m_2) \delta_{12} = s_1 s_2 \delta_{12} (s_{GMT})^2 \tag{9}$$

where $\delta_{12}$ is the pairwise combinatorial effect with respect to the positions of $m_1$ and $m_2$, as computed from M2 matrix as described in "Estimation of Combinatorial Effects" section.

When three mismatches are considered, it is ideal to estimate the combinatorial effect of all three. Unfortunately, the number of possible combinations increases exponentially, which makes experimental estimation of parameters impractical. Here, we consider a sequential model to add the 3rd mismatch, $m_3$, to the model of two mismatches $S(g, m_1, m_2)$:

$$S(g, m_1, m_2, m_3) = S(g, m_1, m_2) S(g, m_3) \delta_{12,3} \tag{10}$$

where $\delta_{12,3}$ is the additional combinatorial effect and is modeled to be the geometric mean of pairwise combinatorial effects of $\delta_{13}$ and $\delta_{23}$.

Combining Eqs. (8)–(10), and the above definition of $\delta_{12,3}$, we have:

$$S(g, m_1, m_2, m_3) = s_1 s_2 s_3 \delta_{12} \sqrt{\delta_{13} \delta_{23}} (s_{GMT})^3 \tag{11}$$

Note that $m_1, m_2, m_3$ are indeed unordered. With a different order in the sequential model, $S(g, m_1, m_2, m_3)$ can also be computed as $s_1 s_2 s_3 \delta_{13} \sqrt{\delta_{12} \delta_{23}} (s_{GMT})^3$ or $s_1 s_2 s_3 \delta_{23} \sqrt{\delta_{12} \delta_{13}} (s_{GMT})^3$. Therefore, we compute $S(g, m_1, m_2, m_3)$ to be the geometric mean of the scores computed with all three possible orders in the sequential model, simplified as follows:

$$S(g, m_1, m_2, m_3) = s_1 s_2 s_3 (\delta_{12} \delta_{13} \delta_{23})^{2/3} (s_{GMT})^3 \tag{12}$$

Finally, we extend the three-mismatch model to $k$ mismatches using mathematical induction approach, and define the MOFF score to be the logarithm

of predicted off-on ratio:

$$S_{MOFF} = \sum_{i=1}^{k} \log(s_i) + \frac{2}{k} \sum_{i=1}^{k} \sum_{j=1}^{i-1} \log(\delta_{ij}) + k \log(s_{GMT}) \tag{13}$$

To assess the genome-wide specificity of a given gRNA, we first mapped the gRNA to the genome to search for potential off-target sites harboring up to 6 mismatches using CRISPRitz, a software for rapid and high-throughput in silico off-target site identification[63]. Next, we defined a MOFF-aggregate score, which is the logarithm of the sum of the MOFF-target scores for all the potential off-target sites. We note that, the current version of MOFF only considers genomic sites with mismatches, but not indels, relative to the 20-nt crRNA sequence as the potential off-targets.

**Evaluation of the models and feature importance analysis.** To evaluate the performance of our model, we curated three independent testing datasets generated by three different platforms, i.e., GUIDE-seq[12], TTISS[18], and CHANGE-seq[19]. GUIDE-seq dataset includes 348 detected off-target sites for 9 gRNAs, TTISS dataset contains 630 detected off-target sites across 59 gRNAs and CHANGE-seq dataset consists of 96,555 detected off-target sites corresponding to 109 gRNAs. For each gRNA-target pair, we measured the off-target effect as off-on ratio, which is calculated as the detected off-target reads divided by on-target reads. For each gRNA, we measured its genome-wide specificity as overall off-on ratio, which is calculated as the total detected off-target reads across the genome divided by on-target reads. We reasoned that the classification of off-target and untargeted sites is highly dependent on the sensitivity of each platform, where the off-target effects indeed take continuous values that may differ by several orders of magnitude. Thus, we adopted the Spearman correlations between measured and predicted off-on ratios for quantitative evaluations.

We further evaluated the importance of different features using Gini importance, which was implemented through the Random Forest Regressor module from the scikit-learn package in Python with default parameters. For MOFF-target, we consider three features: IME, CE, and GMT. For MOFF-aggregate, we consider two features: (1) the mismatch-dependent feature, which is the sum of the predicted mismatch-dependent effects (IME + CE) without considering GMT; (2) the GMT effect corresponding to each gRNA.

**Comparison of off-target prediction methods.** We compared the performance of MOFF to five representative off-target prediction methods: (1) Cutting Frequency Determination (CFD) score is the multiplication of single mismatch effects derived from a cleavage dataset targeting the coding sequence of the human CD33 gene in MOLM-13 cells[9]. For the implementation of CFD, we used the Supplementary Table 19 of their original publication which includes all the single-mismatch effects. (2) Elevation is a two-layer regression model where the first layer learns to predict the effects of single-mismatch and the second layer learns how to combine single-mismatch effects into a final score[24]. The source codes of Elevation were downloaded from https://github.com/Microsoft/Elevation. (3) CNN_std is a deep learning model to predict off-target effects using a standard convolutional neuron network[27] and the source codes to implement CNN_std were downloaded from https://github.com/MichaelLinn/off_target_prediction. (4) CRISPR-Net is a more recent deep learning method using a recurrent convolutional network, which shows superior performance compared to other machine learning approaches[29]. The source codes to implement CRISPR-Net were downloaded from https://codeocean.com/capsule/9553651/tree/v1. And (5) CRISPRoff is an approximate binding energy model for the Cas9-gRNA-DNA complex, which systematically combines the energy parameters obtained for RNA-RNA, DNA-DNA, and RNA-DNA duplexes[31]. The source codes for executing CRISPRoff were downloaded from https://github.com/RTH-tools/crisproff.

We adopted the same strategy to evaluate the performance of different methods as described in "Evaluation of the models and feature importance analysis" section. We note that aggregation models for Elevation and CRISPR-Net assigned different weights to off-targets sites occurring at genic or non-genic regions, which were trained on cell viability screen data of gRNAs targeting non-essential genes; however, cell viability is not a direct indication of DNA cleavage at off-target sites and the genomic features used for training are not associated with the sequence determinants for gRNA specificity. Therefore, the performance of their models degraded when applied to the datasets that directly measure DNA cleavage at off-target sites across the genome (Supplementary Fig. 18). For a fair comparison, we used logarithm of the sum of individual scores for all potential off-target sites identified by CRISPRitz[63] to predict gRNA specificity for all these methods.

**Application to gRNA design.** We collected Avana, GeCKO-v2, and Sanger CRISPR screen data[43,69] from the Depmap portal at https://depmap.org/portal/download. The efficiency of gRNAs was predicted using the DeepHF method[44], and the off-target effect of gRNAs was predicted using MOFF-aggregate. As the analysis involves the computation on tens of thousands of gRNAs, we configured the alignment input to allow up to five mismatches for computational efficiency. All the gRNAs were then classified into three categories: Low efficiency (DeepHF score < 0.6), High efficiency High off-target (DeepHF score > 0.6 and MOFF-

aggregate > 1), and High efficiency Low off-target (DeepHF score > 0.6 and MOFF-aggregate < 1).

To compare the effective size of gRNAs within different categories, we compared the average log-fold change (LFC) of gRNAs targeting 1246 core essential genes and 758 non-essential genes[70] within each category using strictly standardized mean difference (SSMD), which is widely used for quality control in the high-throughput screen data. Higher SSMD values indicate that gRNAs can better discriminate essential and non-essential genes, therefore achieving greater effective size.

To test the cross-library reproducibility for different types of gRNAs, we compared gRNAs targeting 529 cell-specific essential genes from Avana data and Sanger data. Specifically, for gRNAs in different categories, we measured the Pearson correlation between averaged LFC of gRNAs targeting the same genes in Avana and Sanger data across different cell lines. Higher correlation indicates that the effects of gRNAs are more reproducible among different libraries. The cell-specific essential gene list was downloaded from the Depmap portal at https://ndownloader.figshare.com/files/27902064.

**gRNA selection for allele-specific editing.** Given the local DNA sequences of the wildtype and mutant alleles, we first searched for all the possible gRNAs followed by a PAM (NGG) motif targeting the DNA sequence of the mutant allele that harbors a single mutation compared to the wildtype allele. The selected gRNAs, which we termed seed gRNAs, are perfectly matched to the mutant allele, and have a single mismatch relative to the wildtype allele. Next, we introduced all possible single mismatches to the seed gRNAs to generate the candidate gRNAs that differ by 1-nt from the mutant and 2-nt from the wildtype allele. To select the best gRNAs from these candidates, we predicted the MOFF-target scores between the gRNAs and the targeted DNA sequence from the mutant and wildtype alleles, which indicate the sensitivity and selectivity of the gRNAs, respectively. In practice, we selected gRNAs that satisfy: (1) MOFF-target scores at mutant allele >0.5 to ensure high knockout efficiency, (2) the ratio of MOFF-target scores between wildtype allele and mutant allele <0.2, for high selectivity.

**Reporting summary**. Further information on research design is available in the Nature Research Reporting Summary linked to this article.

## Data availability
The raw sequencing data of screens generated from this study are available under NCBI Sequence Read Archive (SRA) (SRA code: PRJNA732904), which can be accessed at https://www.ncbi.nlm.nih.gov/bioproject/PRJNA732904. The raw and processed read counts of the screens generated from this study are provided as Supplementary Data files. Three datasets used in this study are available under NCBI SRA: GUIDE-seq (SRA code: SRP050338), TTISS (SRA code: PRJNA602092), and CHANGE-seq (SRA code: PRJNA625995). Data for Avana, GeCKO-v2 and Sanger CRISPR screens are obtained from the Depmap portal at https://depmap.org/portal/download. Source data are provided with this paper.

## Code availability
The source codes for MOFF software can be accessed from https://github.com/MDhewei/MOFF. To provide a permanent reference to the version of the code used in this study and improve reproducibility, a DOI[71] (https://doi.org/10.5281/zenodo.5792391) has been obtained for this Github repository.

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

## Acknowledgements

We thank Dr. Traver Hart for critical discussion. We thank Dr. Dan Su for sharing the LentiGuide-BSD plasmid. We thank Dr. Jianjun Shen, MDACC-Smithville Molecular Biology Core and MDACC-Smithville Next Generation Sequencing Core for the help of sequencing. This work was supported by the CPRIT grant RR160097 (to H.X.), the CPRIT grant RP170002 (to J.J.S.), the NIH grant R35GM137927 (to H.X.), and the NIH grant R01HL157714 (to X.G.). H.X. is a CPRIT Scholar in Cancer Research.

## Author contributions

H.X., R.F., and W.H. conceptualized the study. R.F. and E.B. performed the experiments. W.H. developed the software. W.H., J.D., O.D.V., C.H., and H.W. performed computational analysis. M.D. performed the biophysical simulation analysis. L.Z., Y.W., and D.M. provided conceptual input. M.D., Y.C., and X.G. helped data interpretation. H.X. supervised the project. All authors participated in writing the manuscript.

## Competing interests

The authors declare no competing interests.
