## [Peer Review File · Nature Communications]

Reviewers' Comments:

Reviewer #1:

Remarks to the Author:

The manuscript by Fu and colleagues present novel approaches to measuring, modeling and predicting off-target cleavage of Cas9/gRNA complexes. Overall the study is solid, scientifically sound and well presented.

The authors present a novel variation of the way others have been doing lentiviral CRISPR screens. The new approach enables for a better and more consistent evaluation of off-target cleavage. In essence they are including an internal control in each experiment to be able to normalize for other variables that can affect the overall cutting rate in these types of experiments. It may be worth more clearly highlighting the importance and advance of this methodology in a head to head analysis of results without use of the internal normalizations (which is what others do). While the results in Figure 2, are nothing completely new, the new experimental system enables the generation of more off-target data than prior studies. This is a critical need in the field. The authors also develop a novel kinetic model that appears to explain guide-intrinsic mismatch tolerance. This model is more complex than previous kinetic models, and performs well when compared to other off-target prediction models although there is still room for improvements. It would be worth expanding on what more is needed to further improve on these models.

Lastly, the authors use their new understanding and model to selectively target specific alleles by taking advantage of the "epistasis-like" combinatorial effect of dual mismatches. For this, they use gRNA with a single mismatch to the specific allele and a double mismatch to the wild-type sequence such that the gRNA only cleaves at the single mismatch allele-specific site. This allows the authors to better differentiate between highly similar sequences. This is really a nice piece to this study. It would be nice to expand the discussion on more generically using non-perfection match gRNAs and an improved understanding of off-target cleavage to improve Cas9 specificity. This is understated in the current manuscript.

Reviewer #2:

Remarks to the Author:

Overall this is a very exciting report, contributing to an emerging consensus that R-loop formation is a critical step in determining Cas9 cleavage potential. The authors use a clever dual target reporter system to evaluate the impact of mismatches on gRNA specificity. They develop a rule-based model based on biophysical principles and empiric data that performs well on predicting off-targets in test datasets. This score could be widely useful for Cas9 off-target prediction which has been a challenging area.

1. My major criticism is that DNA/RNA bulges are not included in their dataset so the prediction is overlooking an important class of off-target sites (e.g. PMID: 24838573). I suggest the authors include a library to test the impact of RNA and DNA bulges and to test if this boosts the performance of the model. If this could not be done, tools that can handle bulges (like CRISTA) may still need to be used for comprehensive off-target evaluation (PMID: 33288953).
2. For allele-specific editing applications, could the authors produce a predictive tool that would input on-target and single mismatch off-target sequence and output single mismatch gRNAs to maximize allele-specific cleavage. Essentially to demonstrate what was done for Fig. 5 could be done prospectively for future allele-specific editing applications. This would be very useful for the editing community.
3. The combinatorial effect estimation of MOFF only considers two mismatches, however, three or more mismatches combination might also contribute to off-target effects. Would considering 3-position combination effect obtain better prediction performance?
4. Since a new high-quality off-target dataset was constructed to develop MOFF, could the authors try a machine learning model on this dataset and compare it with MOFF? For example, training a gradient boosted tree with mismatch types (feature) on numbers of reads of cleavage on off-target (label).
5. Currently off-target sites up to 5 mismatches are tested, but I suggest to also assess MOFF on

scoring off-target sites with up to 6 mismatches, as 6-mismatch off-target sites have been reported by several assays like CIRCLE-Seq and CHANGE-Seq.

6. Are the GUIDE-seq, TTISS, and CHANGE-seq off-targets used for MOFF validation and comparison to other tools orthogonally validated, such as via direct targeted sequencing? As stated, each of these methods has varying levels of sensitivity, but it is important to note that they have varying levels of specificity as well. Not all of the "off-target sites detected by" these methods are necessarily true off-targets because these methods have some rate of false positives. It would seem most fair to only use nominated and subsequently sequence-validated off-targets as ground truth for comparison of MOFF to other methods and exclude any false positives from these methods (nominated but not orthogonally validated).

7. For the dual reporter, are inversions between the two cleavage sites observed? I am surprised there would be such frequent deletions between the two cleavage sites and no inversions.

8. Is the dual reporter confounded by mismatches that may variably influence the likelihood of MMEJ repair (as the two microhomologies become increasingly mismatched)? As I understand it the predicted frequency of deletions observed in the "right cleavage" condition assumes a certain frequency of MMEJ-mediated deletions.

9. For Fig. 2c, suggest to clarify scale for heatmap.

10. For Fig. 2d, is nucleotide frequency at each position for the tested gRNAs distributed roughly evenly? May consider showing as a supplemental figure so it will be clear that this observation would not be affected by any bias in the nucleotide composition of the tested gRNAs?

11. Also for Fig. 2d, could this result also suggest that nucleotides on the same face of the helix as the PAM (adjacent, ~10 nt away, and ~20 nt away, which seem to match the spikes in the plot) are important for GMT?

12. Clarify how the MOFF score formula was developed (log, summation, etc) including reasoning behind definition and other variations explored.

13. Suggest to specify sequencing depth in experiments (quartiles and range).

Response to the reviewers:

We are very grateful to the editor for giving us the opportunity to revise our manuscript. We also thank all the reviewers for their support of the importance of our reported findings, and for their constructive and valuable comments and suggestions. We have performed additional analyses to address all the concerns raised by reviewers and expanded more detailed descriptions and discussions suggested by reviewers to clearly demonstrate the advance of our new dual-target system, MOFF and MOFF-based applications. All the modifications in the current manuscript are highlighted in yellow. Our point-to-point responses to the comments are as follows:

Reviewer #1 (Remarks to the Author):

The manuscript by Fu and colleagues present novel approaches to measuring, modeling and predicting off-target cleavage of Cas9/gRNA complexes. Overall the study is solid, scientifically sound and well presented.

The authors present a novel variation of the way others have been doing lentiviral CRISPR screens. The new approach enables for a better and more consistent evaluation of off-target cleavage. In essence they are including an internal control in each experiment to be able to normalize for other variables that can affect the overall cutting rate in these types of experiments.

It may be worth more clearly highlighting the importance and advance of this methodology in a head to head analysis of results without use of the internal normalizations (which is what others do). While the results in Figure 2, are nothing completely new, the new experimental system enables the generation of more off-target data than prior studies. This is a critical need in the field.

In the revised manuscript, we have modified the description of the dual-target system to highlight the advantage and rationale of this system. On page 4, we added: “Since the off- and on-targets are integrated to the same genomic locus and are PCR-amplified together, the on-target cleavage rate acts as an internal control for the normalization against confounding factors in the experiment. Compared to the single-target design without the use of internal normalizations, the dual-target design is expected to reduce the experimental variations and biases for accurate measurement of off-on ratios”. On page 6, we added: “Overall, these head-to-head comparisons confirmed the rationale and advantage of the dual-target system that improves the accuracy of off-on ratio measurements via internal normalization using on-target controls”.

We have also performed a head-to-head comparison to the previous single-target system focusing on two aspects. First, we measured the correlation of off-on ratios between biological replicates or different barcode sets. As shown in Figure 1g and Supplementary Figure 3, the dual-target system reduced the experimental variations and biases between biological replicates or different barcode sets, supporting the rationale of the new design. Second, we measured the correlation of off-on ratios between high-throughput synthetic system (dual-target or single-target design) and genomic off-target detection methods (GUIDE-seq or WGS) on 296 reported positive off-target sites corresponding to 35 benchmark gRNAs collected from previous *in vitro* (Tsai et al., 2015) or *in vivo* (Anderson et al., 2018) studies. As shown in Figures 1i-1j and Supplementary Figures 4, the off-on ratios detected by the dual-target system showed higher consistency with the measures from the *in vitro* GUIDE-seq and an *in vivo* study on mouse embryos using WGS, as compared to the single-target system. The figure below shows a side-by-side comparison combining the results in Figures 1i-1j and Supplementary Figure 4.

The authors also develop a novel kinetic model that appears to explain guide-intrinsic mismatch tolerance. This model is more complex than previous kinetic models and performs well when compared to other off-target prediction models although there is still room for improvements. It would be worth expanding on what more is needed to further improve on these models.

Considering the recognition and cleavage mechanism from a biophysical viewpoint, Cas9 activity is dictated by the free-energy change during the formation of R-loop, which largely depends on the base-pairing energies. Currently, the kinetic model we used to explain guide-intrinsic mismatch tolerance (GMT) and combinatorial effect is mainly based on the energy barrier caused by the mismatches between gRNA and target DNA sequence during the R-loop formation. The free-energy landscape can also be related to other factors such as the interaction between Cas9 and unwound DNA strand, the conformational change of Cas9-gRNA structure, as well as the folding stability of gRNAs. These factors can also be integrated to improve the performance of the current model. We have modified our manuscript to discuss this on page 14: “In addition to R-loop formation, the interaction between Cas9 and unwound target DNA strand, the conformational change of Cas9-gRNA structure, as well as the folding stability of gRNAs, collectively contribute to the sequence-dependent free-energy landscape. Therefore, the kinetic model can be further improved by taking these factors into consideration, which will lead to a better explanation of off-target effect from a biophysical perspective”.

Lastly, the authors use their new understanding and model to selectively target specific alleles by taking advantage of the "epistasis-like" combinatorial effect of dual mismatches. For this, they use gRNA with a single mismatch to the specific allele and a double mismatch to the wild-type sequence such that the gRNA only cleaves at the single mismatch allele-specific site. This allows the authors to better differentiate between highly similar sequences. This is really a nice piece to this study. It would be nice to expand the discussion on more generically using non-perfection match gRNAs and an improved understanding of off-target cleavage to improve Cas9 specificity. This is understated in the current manuscript.

We have added more discussions on the applications of mismatched gRNAs for the improvement of Cas9 specificity on page 15: “Current strategies for allele-specific editing using discriminating gRNAs are mainly focused on the mutations in the PAM or the seed region. Although several successful applications have been reported using Cas9 and its orthologs/variants (Gyorgy et al., 2019; Kim et al., 2018; Wu et al., 2020), the selectivity of allele-specific editing remains a significant challenge. Based on the “epistasis-like” combinatorial effect of dual mismatches that we reported, we propose to use mismatched gRNAs to improve the selectivity of allele-specific genome editing, where the intended allele harbors a single tolerable mismatch relative to the gRNA and the unintended allele harbors two intolerable mismatches. The high-throughput allele-

editing screen confirmed that a vast majority of cancer hotspot mutations located either in the seed region or non-seed region are selectively targetable through the computer-aided design of mismatched gRNAs. This improvement expands the application domain of allele-specific editing as a time-saving and cost-effective approach for the perturbation of endogenous mutant alleles, thus benefiting functional studies of disease- or trait-associated heterozygous point mutations in a variety of species. Besides allele-specific editing, mismatched gRNAs can be applied to selectively edit one of the paralogous genes that share high degrees of sequence homology, which will facilitate the elucidation of genetic interactions and functional crosstalk between paralogs. Moreover, while our validations are focused on cancer hotspot mutations, the strategy of computer-aided design of mismatched gRNAs holds the clinical potential to correct dominant-negative mutations that drive mendelian disorders, which will be addressed in future work”.

Reviewer #2 (Remarks to the Author):

Overall this is a very exciting report, contributing to an emerging consensus that R-loop formation is a critical step in determining Cas9 cleavage potential. The authors use a clever dual target reporter system to evaluate the impact of mismatches on gRNA specificity. They develop a rule-based model based on biophysical principles and empiric data that performs well on predicting off-targets in test datasets. This score could be widely useful for Cas9 off-target prediction which has been a challenging area.

1. My major criticism is that DNA/RNA bulges are not included in their dataset so the prediction is overlooking an important class of off-target sites (e.g. PMID: 24838573). I suggest the authors include a library to test the impact of RNA and DNA bulges and to test if this boosts the performance of the model. If this could not be done, tools that can handle bulges (like CRISTA) may still need to be used for comprehensive off-target evaluation (PMID: 33288953).

We agree that the lack of analysis of DNA/RNA bulges is a major limitation of MOFF. During the design of our study, we considered including DNA/RNA bulges in initial library design. However, we found that the library size could increase by nearly one order of magnitude due to the combinatorial effect of all possible paired mismatch/bulges. Moreover, although DNA/RNA bulges are an important class of off-target sites, Cas9 has less tolerance to bulges than mismatches, which has been indicated by Doench et al. (Doench et al., 2016) and Jones Jr et al. (Jones et al., 2021). In our independent analysis during research planning, we found that DNA/RNA bulges

account for <10% of the overall detected off-target sites and reads (see table below) across three different datasets from the genome-wide detection of Cas9 off-targets. Due to these considerations, we decided to focus the current study on mismatch type. The DNA/RNA bulges will be included in the future development of MOFF. We hope this is acceptable.

We also modified the discussion part to point out this limitation of our study on page 14: “We envision that MOFF can be further improved by: i) supplying advanced machine learning techniques with known rules and biophysical laws, followed by training the model on a compendium of independent datasets; ii) considering DNA/RNA bulges, which account for ~10% of genomic off-targeting events (Lin et al., 2014); iii) incorporating sequence-independent features such as chromatin structure and epigenetic markers, as reported previously (Kim and Kim, 2018; Lazzarotto et al., 2020)”.

Method	PMID	Sample source	# of off-target sites	# of bulges	% (#: bulge/off-target sites)	Reads of off-target sites	Reads of bulges	% (reads: bulge/off-target sites)	On-target sites
CIRCLE-seq	28459458	HEK293	1253	82	6.54	119676	5192	4.34	HEK-293 site1, site2, site3, site4
		K562	1149	53	4.61	38002	2550	6.71	EMX1, FANCF, RNF2, HEK293 site1, site2, site3
		U2OS	3826	112	2.93	330187	9516	2.88	FANCF, RNF2, VEGFA site2, site3
		U2OS_R1	620	13	2.10	54086	344	0.64	EMX1, VEGFA site1
		U2OS_R2	508	10	1.97	25062	170	0.68	EMX1, VEGFA site1
CHANGE-seq	32541958	human primary CD4+/CD8+ T	201934	10514	5.21	35685348	2120605	5.94	109 sites
GUIDE-seq (corrected*)	25513782 29036168	HEK293/U2OS	403	73	18.11	87915	5929	6.74	EMX1, FANCF, RNF2, HEK293 site1, site2, site3, site4, VEGFA
corrected*: calculation based on the table with pairwise alignment to account for bulges (PMID: 29036168), the original table only has one off-target with bulge (PMID: 25513782)									

2. For allele-specific editing applications, could the authors produce a predictive tool that would input on-target and single mismatch off-target sequence and output single mismatch gRNAs to maximize allele-specific cleavage. Essentially to demonstrate what was done for Fig. 5 could be done prospectively for future allele-specific editing applications. This would be very useful for the editing community.

We have added a module to MOFF software, named MOFF-allele. Given the local DNA sequences of the wildtype and mutant alleles, MOFF-allele first searches for all the possible gRNAs followed by a PAM (NGG) motif targeting the DNA sequence of the mutant allele that harbors a single mutation compared to the wildtype allele. Next, MOFF-allele introduces all possible single mismatches to the selected gRNAs to generate the candidate gRNAs that differ by 1-nt from the mutant and 2-nt from the wildtype allele. For all the candidates, MOFF-allele

computes the MOFF-target scores between the gRNAs and the targeted DNA sequence from the mutant and wildtype alleles, which indicate the sensitivity and selectivity of the gRNAs for users' selection. The whole package of MOFF including MOFF-target, MOFF-aggregate and MOFF-allele is available at <https://github.com/MDhewei/MOFF>.

In the main text, we have added a sentence to introduce MOFF-allele to the users on page 13: "To address the needs of guide design in allele-specific editing, we implemented a module in the MOFF package, named MOFF-allele, that allows the users to select the optimal mismatched gRNA for their study".

3. The combinatorial effect estimation of MOFF only considers two mismatches, however, three or more mismatches combination might also contribute to off-target effects. Would considering 3-position combination effect obtain better prediction performance?

We agree that 3+ mismatch combinations may contribute to off-target effects beyond the model of paired combination in our study. Ideally, all the combinations of 3+ mismatches should be considered. However, the number of combinations increases exponentially, and it is impossible to estimate the parameters of all 3+ combinations from experiments. Therefore, we used a mathematic inductive approach to extend the paired combinations to the cases of 3+ mismatches. We have added detailed explanations of our method and the MOFF scoring function in the section of "Model-based off-target prediction with MOFF" in Online Methods (page 29). In this section, we explained why we used a mathematical inductive approach, instead of wet experiments, for the estimation of off-target effect with 3+ mismatches.

4. Since a new high-quality off-target dataset was constructed to develop MOFF, could the authors try a machine learning model on this dataset and compare it with MOFF? For example, training a gradient boosted tree with mismatch types (feature) on numbers of reads of cleavage on off-target (label).

To address this question, we tested three different machine learning models: a Gradient Boosted Tree (XGBoost) as suggested by the reviewer, a Random Forest Regressor (RF), and a Support Vector Machine (SVM). A consistent encoding method storing on- to off-target sequence nucleotides at each position was used as input. The models were trained on the same dataset as MOFF, and validated on third-party datasets from CHANGE-Seq, GUIDE-Seq, and TTISS. The performance was evaluated as the spearman correlation between predicted off-target effects and

measured off-on ratios in the test datasets. Because we designed 1-3 mismatch off-target sites in our library, the test datasets contain the off-target sites with 1-3 mismatches only. As shown in Supplementary Figure 15, the rule-based method (MOFF) performed consistently better than the machine learning-based methods. This result supports our argument that complex models are sensitive to platform-specific biases and are prone to overfitting, whereas rule-based approaches such as MOFF are more robust against the variation of platforms. We extend our discussion to address this point on page 14.

5. Currently off-target sites up to 5 mismatches are tested, but I suggest to also assess MOFF on scoring off-target sites with up to 6 mismatches, as 6-mismatch off-target sites have been reported by several assays like CIRCLE-Seq and CHANGE-Seq.

MOFF supports the assessment of off-target effects on sequences with up to 6 mismatches. The description in the original manuscript is incorrect. Indeed, we used up to 6 mismatches for the assessment of MOFF-target and MOFF-aggregate on the datasets of Guide-seq, CHANGE-seq, and TTISS (Figures 4b-4c and Supplementary Table 12). Only when we tested MOFF on genome-wide screening datasets, we used up to 5 mismatches (Figures 4d-4e and Supplementary Figures 12-13). This is because the analysis involves tens of thousands of gRNAs, we adjusted the threshold for computational efficiency. We believed that, for users who are interested in the optimization of gRNA library design using MOFF, up to 5 mismatches is a reasonable configuration to balance the accuracy and computational cost. We have modified the main text and Online Methods section to clarify this issue.

6. Are the GUIDE-seq, TTISS, and CHANGE-seq off-targets used for MOFF validation and comparison to other tools orthogonally validated, such as via direct targeted sequencing? As stated, each of these methods has varying levels of sensitivity, but it is important to note that they have varying levels of specificity as well. Not all of the “off-target sites detected by” these methods are necessarily true off-targets because these methods have some rate of false positives. It would seem most fair to only use nominated and subsequently sequence-validated off-targets as ground truth for comparison of MOFF to other methods and exclude any false positives from these methods (nominated but not orthogonally validated).

We agree that each method to detect off-targets has a certain rate of false positives. Using nominated and validated bona fide off-targets for comparison of MOFF to other methods would be fairer. Two of three datasets, GUIDE-seq and CHANGE-seq, are orthogonally validated via anchored multiplex PCR (AMP)-based next-generation sequencing and targeted tag integration sequencing, respectively. Unfortunately, we found that the validated dataset was not accessible for GUIDE-seq. Therefore, we compared different methods on 204 positive sites (excluding bulges) that are orthogonally validated in CHANGE-seq dataset. The result suggests that MOFF outperformed other methods, as shown in the figure below.

7. For the dual reporter, are inversions between the two cleavage sites observed? I am surprised there would be such frequent deletions between the two cleavage sites and no inversions.

Yes, we have observed rare cases where two cleavage sites are inverted. Among all the sequenced reads, there is a very small proportion (~0.005%) of edited reads associated with inversions between two cleavage sites. Thus, the impact of conversion on the accuracy of our measures is insignificant.

Our dual-target design could lead to a 35-bp DNA fragment under “both cleavage” condition. Li et al. (Li et al., 2015) showed that leveraging Cas9 with a pair of gRNAs could generate inversions of DNA fragments ranging in size from three dozens of bp to hundreds of kb in human cells. They reported inversion rates of <1% to ~30% for DNA fragments of size 700bp to 80,000bp, but didn’t report the actual rate for smaller DNA fragments of size <100bp. Binda et al. (Binda et al., 2020) used dual gRNAs (gGag1 and gGag3) to attack HIV DNA, of which the DNA fragment between two Cas9 cleavage sites is 80bp. They reported that the frequency of inversions was too low to be detected. When they further designed another two pairs of gRNAs (gGag1 and gTatRev,

gGag1 and gEnv2) to target DNA fragments in sizes of thousands of bps, 1%-7% of inversion rate was estimated, in consistency with Li et al. results. Combining our analysis and these previous reports, it seems that the inversion rate of small fragments with size <100bp is low. This could be due to a complex process involving CRISPR targeting and DSB repair in the local region, of which the mechanism is yet unclear.

8. Is the dual reporter confounded by mismatches that may variably influence the likelihood of MMEJ repair (as the two microhomologies become increasingly mismatched)? As I understand it the predicted frequency of deletions observed in the “right cleavage” condition assumes a certain frequency of MMEJ-mediated deletions.

Thank you for raising this issue. The number of mismatches indeed can influence the likelihood of MMEJ repair. Due to MMEJ, some “cleavage at off-target only” cutting events lead to large deletion, which can be wrongly assigned to the “both cleavage” state. In the transition matrix in Figure 1f, we provide an estimation of MMEJ rate of ~25%. Because the estimation is based on perfect-matched gRNA and target in the control sequences, and there is a lower MMEJ rate when mismatches occur in microhomology sequences, this estimation is an upper-bound of the actual MMEJ rate in mismatched sequence. Therefore, we anticipate a variable MMEJ rate between 0.0–0.25 when different mismatches are concerned.

How does the variation of MMEJ rate impact the accuracy of off-on ratio calculation? We consider the formula for off-on-ratio calculation as follows:

$$\text{ratio} = \frac{s_1 + s_3}{s_2 + s_3}$$

Where s_1 , s_2 and s_3 are the estimated read counts for “cleavage at off-target only”, “cleavage at on-target only”, and “both cleavage”, respectively. The figure below shows the distributions of read counts in s_1 , s_2 and s_3 , where s_2 is much larger than s_1 and s_3 . Since the MMEJ affects the assignment of reads to “cleavage at off-target only” (s_1) or “both cleavage” (s_3) (Figure 1f), the numerator (sum of s_1 and s_3) is unchanged with different MMEJ rates. Because s_2 is much larger than s_3 and stabilizes the denominator, the variation of s_3 due to MMEJ only contributes a small fraction to the total variation of the ratio. Taking the median value of s_1 , s_2 and s_3 for estimation, we anticipate a variation of 0.2% of off-on ratio, given the variation of 0.0-0.25 in the MMEJ rate. Therefore, although the variation of MMEJ rate results in a confounding factor to the computation of off-on ratio, its impact is insignificant and tolerable.

9. For Fig. 2c, suggest to clarify scale for heatmap.

We have added the scale clarification for the heatmap in Figure 2c.

10. For Fig. 2d, is nucleotide frequency at each position for the tested gRNAs distributed roughly evenly? May consider showing as a supplemental figure so it will be clear that this observation would not be affected by any bias in the nucleotide composition of the tested gRNAs?

We have added the nucleotide frequency at each position for the tested gRNAs in Supplementary Figure 16. As shown in the figure, four types of nucleotides are almost evenly distributed at each position of gRNA.

11. Also for Fig. 2d, could this result also suggest that nucleotides on the same face of the helix as the PAM (adjacent, ~10 nt away, and ~20 nt away, which seem to match the spikes in the plot) are important for GMT?

Yes, we also observed this phenomenon and agree with the reviewer that the result probably suggests an association between GMT effect and the DNA helix structure starting from PAM. The mechanism underlying this observation remains evasive, therefore we didn't highlight it in the manuscript. It would be interesting to collaborate with biophysicists and structural biologists to further explore the mechanism. We hope it is acceptable to leave this study to the future.

12. Clarify how the MOFF score formula was developed (log, summation, etc) including reasoning behind definition and other variations explored.

We have added detailed explanations of the MOFF score in the section of “Model-based off-target prediction with MOFF” in Online Methods (page 29).

13. Suggest to specify sequencing depth in experiments (quartiles and range).

We have plotted histograms to show distributions of the number of sequenced reads per gRNA-target pair, which represent the sequencing depths of the experiments with dual-target design. These results were added as Supplementary Figure 17.

Reference

- Anderson, K.R., Haeussler, M., Watanabe, C., Janakiraman, V., Lund, J., Modrusan, Z., Stinson, J., Bei, Q., Buechler, A., Yu, C., *et al.* (2018). CRISPR off-target analysis in genetically engineered rats and mice. *Nat Methods* 15, 512-514.
- Binda, C.S., Klaver, B., Berkhout, B., and Das, A.T. (2020). CRISPR-Cas9 Dual-gRNA Attack Causes Mutation, Excision and Inversion of the HIV-1 Proviral DNA. *Viruses* 12.
- Doench, J.G., Fusi, N., Sullender, M., Hegde, M., Vaimberg, E.W., Donovan, K.F., Smith, I., Tothova, Z., Wilen, C., Orchard, R., *et al.* (2016). Optimized sgRNA design to maximize activity and minimize off-target effects of CRISPR-Cas9. *Nat Biotechnol* 34, 184-191.
- Gyorgy, B., Nist-Lund, C., Pan, B., Asai, Y., Karavitaki, K.D., Kleinstiver, B.P., Garcia, S.P., Zaborowski, M.P., Solanes, P., Spataro, S., *et al.* (2019). Allele-specific gene editing prevents deafness in a model of dominant progressive hearing loss. *Nat Med* 25, 1123-1130.
- Jones, S.K., Jr., Hawkins, J.A., Johnson, N.V., Jung, C., Hu, K., Rybarski, J.R., Chen, J.S., Doudna, J.A., Press, W.H., and Finkelstein, I.J. (2021). Massively parallel kinetic profiling of natural and engineered CRISPR nucleases. *Nat Biotechnol* 39, 84-93.
- Kim, D., and Kim, J.S. (2018). DIG-seq: a genome-wide CRISPR off-target profiling method using chromatin DNA. *Genome Res* 28, 1894-1900.
- Kim, W., Lee, S., Kim, H.S., Song, M., Cha, Y.H., Kim, Y.H., Shin, J., Lee, E.S., Joo, Y., Song, J.J., *et al.* (2018). Targeting mutant KRAS with CRISPR-Cas9 controls tumor growth. *Genome Res* 28, 374-382.
- Lazarotto, C.R., Malinin, N.L., Li, Y., Zhang, R., Yang, Y., Lee, G., Cowley, E., He, Y., Lan, X., Jividen, K., *et al.* (2020). CHANGE-seq reveals genetic and epigenetic effects on CRISPR-Cas9 genome-wide activity. *Nat Biotechnol* 38, 1317-1327.
- Li, J., Shou, J., Guo, Y., Tang, Y., Wu, Y., Jia, Z., Zhai, Y., Chen, Z., Xu, Q., and Wu, Q. (2015). Efficient inversions and duplications of mammalian regulatory DNA elements and gene clusters by CRISPR/Cas9. *J Mol Cell Biol* 7, 284-298.
- Lin, Y., Cradick, T.J., Brown, M.T., Deshmukh, H., Ranjan, P., Sarode, N., Wile, B.M., Vertino, P.M., Stewart, F.J., and Bao, G. (2014). CRISPR/Cas9 systems have off-target activity with insertions or deletions between target DNA and guide RNA sequences. *Nucleic Acids Res* 42, 7473-7485.
- Tsai, S.Q., Zheng, Z., Nguyen, N.T., Liebers, M., Topkar, V.V., Thapar, V., Wyvekens, N., Khayter, C., Iafrate, A.J., Le, L.P., *et al.* (2015). GUIDE-seq enables genome-wide profiling of off-target cleavage by CRISPR-Cas nucleases. *Nat Biotechnol* 33, 187-197.

Wu, J., Tang, B., and Tang, Y. (2020). Allele-specific genome targeting in the development of precision medicine. *Theranostics* *10*, 3118-3137.

Reviewers' Comments:

Reviewer #1:

Remarks to the Author:

Fu et al report on novel and improved methods to measure, model and predict off-target cleavage of Cas9/gRNA complexes. The authors have significantly updated the manuscript and have addressed all of my major concerns, the work is a solid addition the field. Specifically, they have added a head to head comparison with an alternative approach to further validate the advance, they have also improved the treatment of where this advance fits and how it can inform future efforts. Specifically discussing the limitations of the current approach and challenges to approaches/models which incorporate more extensive variables. They have adequately addressed many concerns brought up by the second reviewer (in my opinion) , which while valid and important limitations to discuss, present significant challenges and advances beyond the scope of the current study. I have no remaining concerns regarding publication.

Reviewer #2:

Remarks to the Author:

The authors have prepared a thorough revision and have addressed all my concerns either through new analyses or by describing limitations and future directions in the text. The allele-specific editing prediction module is a nice added feature.

I suggest information from the response about inversions could be added briefly to the text since this would be useful for readers to understand.

One additional limitation that I think the authors should note is that the current method does not consider mismatches in the PAM sequence.

Response to the reviewers:

We thank all the reviewers for their agreements and supports of our previous point-to-point responses. As Reviewer #1 has no more concerns regarding our manuscript, we further modified our manuscript conditional on the suggestions raised by Reviewer #2. All the modifications in the current manuscript are highlighted in yellow. Our point-to-point responses to comments of Reviewer #2 are as follows:

Reviewer #1 (Remarks to the Author):

Fu et al report on novel and improved methods to measure, model and predict off-target cleavage of Cas9/gRNA complexes. The authors have significantly updated the manuscript and have addressed all of my major concerns, the work is a solid addition the field. Specifically, they have added a head to head comparison with an alternative approach to further validate the advance, they have also improved the treatment of where this advance fits and how it can inform future efforts. Specifically discussing the limitations of the current approach and challenges to approaches/models which incorporate more extensive variables. They have adequately addressed many concerns brought up by the second reviewer (in my opinion), which while valid and important limitations to discuss, present significant challenges and advances beyond the scope of the current study. I have no remaining concerns regarding publication.

Reviewer #2 (Remarks to the Author):

The authors have prepared a thorough revision and have addressed all my concerns either through new analyses or by describing limitations and future directions in the text. The allele-specific editing prediction module is a nice added feature.

I suggest information from the response about inversions could be added briefly to the text since this would be useful for readers to understand.

It is a good suggestion to mention the rate of inversion induced by double cleavage of Cas9 in our manuscript. We have added the information related to inversions on page 24: "Of note, we only observed rare cases (~0.005%) where two cleavage sites are inverted in our dual-target design (Li et al., 2015). This is consistent with previous report where inversion rate of small DNA

fragments with size <100bp is much lower compared to that of larger fragments of hundreds to thousands bps (Binda et al., 2020). Thus, here we did not include this inversion type in the classification of indel types”.

One additional limitation that I think the authors should note is that the current method does not consider mismatches in the PAM sequence.

Thank you for raising this limitation of our method. We have modified the discussion part to point out this limitation of our study on page 14: “We envision that MOFF can be further improved by: i) supplying advanced machine learning techniques with known rules and biophysical laws, followed by training the model on a compendium of independent datasets; ii) considering DNA/RNA bulges, which account for ~10% of genomic off-targeting events; iii) accounting the mismatches in the PAM sequence, as some alternative PAM sequences, such as NAG, also lead to active Cas9/gRNA editing (Doench et al., 2016); and iv) incorporating sequence-independent features such as chromatin structure and epigenetic markers, as reported previously”.

Reference

Binda, C.S., Klaver, B., Berkhout, B., and Das, A.T. (2020). CRISPR-Cas9 Dual-gRNA Attack Causes Mutation, Excision and Inversion of the HIV-1 Proviral DNA. *Viruses* 12.

Doench, J.G., Fusi, N., Sullender, M., Hegde, M., Vaimberg, E.W., Donovan, K.F., Smith, I., Tothova, Z., Wilen, C., Orchard, R., *et al.* (2016). Optimized sgRNA design to maximize activity and minimize off-target effects of CRISPR-Cas9. *Nat Biotechnol* 34, 184-191.

Li, J., Shou, J., Guo, Y., Tang, Y., Wu, Y., Jia, Z., Zhai, Y., Chen, Z., Xu, Q., and Wu, Q. (2015). Efficient inversions and duplications of mammalian regulatory DNA elements and gene clusters by CRISPR/Cas9. *J Mol Cell Biol* 7, 284-298.